# The global water resources and use model WaterGAP v2.2e: description and evaluation of modifications and new features

Hannes Müller Schmied[1,2], Tim Trautmann[1], Sebastian Ackermann[1], Denise Cáceres[1], Martina Flörke[3], Helena Gerdener[4], Ellen Kynast[5], Thedini Asali Peiris[1], Leonie Schiebener[1], Maike Schumacher[6], and Petra Döll[1,2]

[1]Institute of Physical Geography, Goethe University Frankfurt, Frankfurt am Main, Germany
[2]Senckenberg Leibniz Biodiversity and Climate Research Centre (SBiK-F), Frankfurt am Main, Germany
[3]Engineering Hydrology and Water Resources Management, Ruhr-University of Bochum, Bochum, Germany
[4]Institute of Geodesy and Geoinformation, University of Bonn, Bonn, Germany
[5]Center for Environmental Systems Research, University of Kassel, Kassel, Germany
[6]Geodesy Group, Department of Planning, Aalborg University, Denmark

Correspondence: hannes.mueller.schmied@em.uni-frankfurt.de

**Abstract.** Water - Global Assessment and Prognosis (WaterGAP) is a modeling approach for quantifying water resources and water use for all land areas of the Earth that has served science and society since 1996. In this paper, the refinements, new algorithms and new data of the most recent model version v2.2e are described, together with a thorough evaluation of simulated water use, streamflow and terrestrial water storage anomaly against observation data. WaterGAP v2.2e improves the handling of
inland sinks and now excludes not only large but also small man-made reservoirs when simulating naturalized conditions. The reservoir and non-irrigation water use data were updated. In addition, the model was calibrated against an updated and extended dataset of streamflow observations at 1509 gauging stations. The modifications resulted in a small decrease of the estimated global renewable water resources. The model can now be started using pre-scribed water storages and other conditions, which facilitates data assimilation as well as near real-time monitoring and forecast simulations. For specific applications, the model
can consider the output of a glacier model, approximate the effect of rising $CO_2$ concentrations on evapotranspiration or calculate the water temperature in rivers. In the paper, the publicly available standard model output is described and caveats of the model version are provided alongside the description of the model setup in the ISIMIP3 framework.

## 1   Introduction

The quantitative assessment of global water resources and their use helps to increase our understanding of the freshwater
cycle and supports decision-making. Global hydrological modeling approaches have been developed since the 1990s and one of the pioneers in this field is the global water resources and water use model WaterGAP (Water - Global Assessment and Prognosis) (Alcamo et al., 2003; Döll et al., 2003). To continue to answer relevant scientific and societal questions, such a model system needs to be state-of-the-art in terms of process representation and the databases used. Moreover, informative descriptions of specific model versions are required and are increasingly supplied in global hydrological modeling (Burek
et al., 2020; Hanasaki et al., 2018; Stacke and Hagemann, 2021; Clark et al., 2011; Best et al., 2011; Mathison et al., 2023;

Yokohata et al., 2020), especially when the models are part of model intercomparison exercises. This paper describes the changes to WaterGAP 2 (from now referred to as WaterGAP) from version 2.2d (v2.2d) (Müller Schmied et al., 2021) to the most recent model version 2.2e (v2.2e) to present the modifications and extensions rather than a thorough description of the whole WaterGAP model. Furthermore, it provides a model evaluation against independent data for different model variants and explains its application in the Inter-Sectoral Impact Model Intercomparison Project phase 3 (ISIMIP3) framework (https://www.isimip.org/protocol/3/). While this paper does not repeat the full model overview provided in Müller Schmied et al. (2021), the main characteristics of the model system are described in the next paragraphs, followed by the motivation and rationale of new features of model version v2.2e.

WaterGAP was developed to quantify global-scale water resources as well as water stress with focus on direct human impacts on the natural water cycle by human water use and artificial reservoirs. The model framework (Fig. 1) consists of sectoral water use models that are linked in a submodel (GSWSUSE) to calculate potential net water abstractions from surface water bodies and from groundwater. The computed net abstractions are an input for the WaterGAP Global Hydrology Model that calculates the water storages and fluxes and routes the streamflow to the basin outlet (Fig. 1). WaterGAP as described here operates with a spatial resolution of $0.5°$ x $0.5°$ and at daily time steps.

A model like WaterGAP is used to answer questions with numerical experiments where the model is driven by alternative inputs, for example climate data to quantify the impact of climate change on water resources, or run with different setups or algorithms. One extensively performed experiment is to switch off human water use and artificial reservoirs to evaluate these direct human impacts on the water cycle (e.g. Döll et al., 2020). For this evaluation, WaterGAP is run both in its standard mode ("ant", including direct human impacts) and in a naturalized mode ("nat"), simulating naturalized water flows and storages that would occur if there were neither human water use nor artificial reservoirs/regulated lakes. In model version v2.2d, the naturalized mode assumes that human water use is zero worldwide; "global" reservoirs, which are handled with the reservoir algorithm (storage capacity larger than $0.5$ $km^3$), do not exist and regulated lakes are treated as the original natural lakes. However, in v2.2d, the more than 5000 small reservoirs with storage capacities below $0.5$ $km^3$ are included in the "local lakes" input data (Müller Schmied et al., 2021, their Sect. 4.6) and are still included even in naturalized mode such that evapotranspiration and surface water body storage is overestimated. To avoid this misrepresentation of naturalized condition, the preparation of a specific "local lake" input data set for naturalized runs is required that does not contain the small reservoirs.

The capability of WaterGAP to assess the impact of climate change on the freshwater system is limited, as is the case for most hydrological models, by not being able to simulate the response of vegetation to climate change and an increased atmospheric $CO_2$ concentration. The simulation of vegetation responses (instead of assuming no changes in vegetation that affect evapotranspiration) may result in substantial differences in estimated climate change impacts, for example on groundwater recharge (Reinecke et al., 2021). However, the simulation of vegetation responses is complex and uncertain and a simplified approach is required. Applying the results of Milly and Dunne (2016), who analyzed future evapotranspiration changes of an ensemble of global climate models, we developed an alternative method for calculating potential evapotranspiration under climate change applicable to the Priestley-Taylor PET method. This model variant can be used in an ensemble together with the standard model to approximate the range of uncertainty of future evapotranspiration and runoff changes.

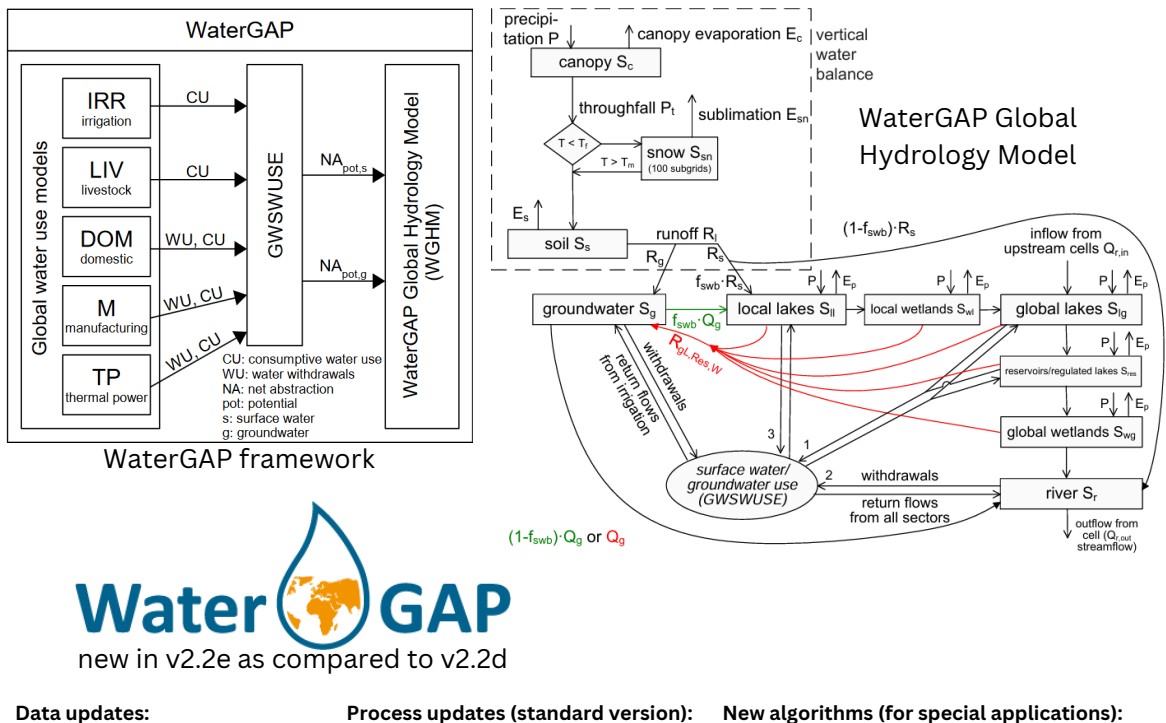

**Figure 1.** Schematics of the WaterGAP framework and the WaterGAP Global Hydrology Model (both taken from Müller Schmied et al., 2021) and summary of data updates, process updates and new algorithms.

Glaciers play a crucial role in the global water cycle (Scanlon et al., 2023; An et al., 2021) but are represented in very few global hydrological models (Telteu et al., 2021). Neglecting the dynamics of water storage in glaciers results in a missing component of the terrestrial water storage and hinders quantifying the impact of glacier mass loss on water resources and sea-level rise. We had developed a glacier component (HYOGA) for a previous version of WaterGAP (Hirabayashi et al.,
2010), which, however, is no longer state-of-the-art. Hence, to enable an optimal consideration of glacier water dynamics, it is preferable to include the output of a dedicated glacier model in a global hydrological model (Hanus et al., 2024; Wiersma et al., 2022). This approach has been implemented in WaterGAP v2.2e, but not in its standard version due to the limited temporal extent of the glacier model output.

An important indicator of water quality is water temperature, especially in a changing climate (Hannah and Garner, 2015;
Van Vliet et al., 2013). Therefore, the Inter-Sectoral Impact Model Intercomparison Project (ISIMIP) has included river water temperature as a requested variable in its recent phase 3. Moreover, the new ISIMIP sector water quality has been formed

that has identified water temperature as one of the essential elements (https://protocol.isimip.org/#/ISIMIP3a/water_quality). Furthermore, the calculation of water temperature helps to assess the heat uptake of inland waters (Vanderkelen et al., 2020). Hence, in WaterGAP v2.2e a simple algorith to calculate the water temperatures of rivers and surface water bodies was intro-
duced.

An important rationale for developing a new model version is to update the input data basis to reflect the current state of the art. To optimally take into account reservoirs in WaterGAP and to be consistent with other global hydrological models participating in the model intercomparison project ISIMIP, it has been necessary to update the reservoirs and regulated lakes data to GRanD (Lehner et al., 2011) version 1.3 and include some additional reservoirs from other sources. In terms of non-
irrigation water use data, two errors (one error in downscaling national level to grid cell level, and one copy-paste error) appeared in WaterGAP v2.2d in creating the domestic water use time series which was subject to be corrected in v2.2e. Furthermore, input data to temporally extend the time series for thermal electricity (from 2010 to 2017) and manufacturing water use (from 2010 to 2016) were available.

Models and their inputs are imperfect and calibration can help to reduce the uncertainty of model output (e.g. Döll et al.,
2016). Hence, WaterGAP has been calibrated against observed mean annual streamflow in a simple but basin-specific manner since its first described version (Alcamo et al., 2003; Döll et al., 2003). With this approach, the bias of simulated streamflow is strongly reduced. Therefore, inclusion of newly available streamflow data in the calibration process is beneficial.

The improvement of already implemented algorithms is another motivation of developing a new model version. Focused groundwater recharge below surface water bodies in (semi)arid grid cells was a feature introduced in WaterGAP v2.2a (Döll
et al., 2014). A modification in WaterGAP v2.2d regarding the handling on grid cells without outflow of liquid water, i.e. internal or sinks, has led to unrealistically high values of groundwater recharge in these cells that are difficult to interpret in a water balance approach, especially when assessing the impact of climate change on groundwater resources (Reinecke et al., 2021). A good example is the Okavango Delta in Botswana, an endorheic basin with a surface water body. Here, approx. 95% of the inflowing water is evaporated, rather than recharge the groundwater (Milzow et al., 2009), while the v2.2d model version
computes very large focused groundwater recharge under the delta. In addition, the modification to handle inland sinks in v2.2d just like any other grid cell has led to outputting a value for streamflow out of the inland sink, which does not reflect reality. Both issues motivate a modification of the handling of inland sinks in the model.

Data assimilation, which requires regular updating of the model states (water storages), was not possible with the standard version of v2.2d as the simulation could not be stopped at a certain point in time (e.g. March 31, 2004) and restarted to continue
the computation (for April 1, 2004) with prescribed initial conditions that have been written out at the end of the previous model run. Therefore, the WaterGAP Global Hydrology Model was modified to enable a monthly restart and successfully applied in data assimilation (Gerdener et al., 2023; Döll et al., 2024). In addition, the restart capability is a prerequisite to applying WaterGAP in water resources monitoring and ensemble forecasts of water resources. Also, it reduces model run times, in particular in climate change assessments. The participation of the model in the ISIMIP3b simulation round requires model
runs for different time periods (e.g. the pre-industrial period starting in the year 1601, the historical time period in the year 1850, and the future in the year 2015). With v2.2d, each run for the future time period would require a transient run with a start

in 1601 to reach full consistency especially between the time periods, leading to a high demand for computing resources and runtime. To perform the multiple scenario evaluation for the 86 years 2015-2100, starting in 1601 would lead to a runtime of 25 hours, while runtime would be only 4 hours if the model could start with prescribed initial conditions in 2015.

To address these scientific demands, WaterGAP was updated to version v2.2e. The objective of this paper is to clearly describe the modifications and new options implemented in WaterGAP v2.2e and to evaluate the impact of the modifications on model results. The paper describes

- the removal of small reservoirs from local lake storage compartment to achieve an improved simulation of naturalized conditions (Sect. 2.1)

– the updated database for reservoirs and regulated lakes (Sect. 2.2)

- the updated and bug-fixed non-irrigation water use data (Sect. 2.3)

- the updated streamflow observations data set used for model calibration (Sect. 2.4)

- the new handling of inland sinks (Sect. 2.5)

- the integration of an alternative approach for PET to improve climate change impact assessments (Sect. 3.1)

– the integration of outputs from a global glacier model (Sect. 3.2)

- the implementation of water temperature calculation (Sect. 3.3)

- the model restart capability (Sect. 3.4)

The remaining paper is organized as follows: Modifications of algorithms and data that affect standard model runs are described in Sect. 2. New options for applications in specific cases are explained in Sect. 3. The model setup and the climate

input data used for this paper are described in Sect. 4. The effects of the modifications for the standard runs are shown in Sect. 5 and for the specific options in Sect. 6. The comparison of model outputs to observations and reference data follows in Sect. 7. A discussion about benefits and limitations of the calibration approach follows in Sect. 8. The standard model output as well as caveats are described in Sect. 9 and Sect. 10, respectively. WaterGAP v2.2e is applied in the Inter-Sectoral Impact Model Intercomparison Project phase 3 (ISIMIP3). The specifics of the model runs and deviations from the ISIMIP model protocol

are described in Sect. 11. The paper ends with the conclusions and outlook in Sect. 12.

This new version integrates, next to technical modifications (Appendix A) several improvements in characterizing human interference in the global water cycle.

## 2 Modifications of algorithms and data affecting standard model results

### 2.1 Naturalized runs: Small reservoirs are no longer considered in naturalized runs

In WaterGAP v2.2d, small reservoirs (<0.5 $km^3$ storage capacity) are simulated as local lakes, whether or not WaterGAP is run in "nat" mode. In WaterGAP v2.2e, the small reservoirs are removed from local lakes in "nat" runs, decreasing the grid cell-specific area share covered by surface water bodies that are simulated with the local lakes algorithm. In standard ("ant") runs, small reservoirs continue to be treated like natural lakes. After integration of updates and new reservoirs from the Global Reservoir and Dam Database (GRanD) 1.3 (Lehner et al., 2011) (Sect. 2.2), there are 5722 small reservoirs with a maximum storage capacity of less than 0.5 $km^3$ in WaterGAP v2.2e. They cover a total maximum area of 31,630 $km^2$.

### 2.2 Reservoir and regulated lake data: GRanD 1.3 integration

In WaterGAP, reservoirs with a storage capacity of at least 0.5 $km^3$ are simulated as so-called "global" reservoirs that receive inflow from the upstream grid cell. Their dynamics are simulated with a filling and operational scheme depending on their main use (irrigation or non-irrigation) (Müller Schmied et al., 2021). Changes to reservoirs and new reservoirs from GRanD (Lehner et al., 2011) version 1.3 together with four additional reservoirs from a preliminary version of the GeoDAR dataset (Wang et al., 2022) were implemented in WaterGAP v2.2e. Reservoirs with a commissioning year until 2020 were selected and mapped to the river network of WaterGAP DDM30 (Döll and Lehner, 2002; Schewe and Müller Schmied, 2022). The location of the new reservoirs was manually co-registered in the drainage network with the help of web-based map information in order to match the given hydrological situation, in particular if a reservoir is located on the mainstream or its tributary. The total number of implemented reservoirs with a storage capacity of at least 0.5 $km^3$ increased from 1082 in WaterGAP v2.2d to 1255 in WaterGAP v2.2e, and the number of regulated lakes increased from 85 to 88. The total maximum storage capacity of the global reservoirs sums up to 5672 $km^3$.

Furthermore, parameters (i.e., commissioning year, assigned outflow cell, etc.) from 12 reservoirs were changed either due to changes from GRanD 1.1 to 1.3 or for correcting flawed parameterization. Multiple reservoirs and regulated lakes may have their outflow cell in the same grid cell. In such cases, they are simulated as one big reservoir or regulated lake, by adding up their maximum area and storage capacity and assigning to this new water body the type (reservoir or regulated lake) and the commissioning year of the actual reservoir or regulated lake with the largest water storage capacity. Thus, for example, a regulated lake and a reservoir can become one reservoir in WaterGAP. Therefore, WaterGAP v2.2e simulates explicitly only a maximum of 1181 reservoirs and 86 regulated lakes (corresponding data available from Müller Schmied and Trautmann (2023)). In addition to these global reservoirs, local reservoirs with a storage capacity smaller than 0.5 $km^3$ were updated to GRanD version 1.3 (Sect. 2.1).

## 2.3 Water use data: Updated non-irrigation water use data

In WaterGAP, domestic water use is calculated on a national level and then downscaled to the grid cells according to the population number per grid cell. Additional information, such as the ratio of rural to urban population per grid cell and the share of the population with access to safe water supply are considered (Flörke et al., 2013). In the 22d version, an error occurred for a few countries in the downscaling procedure because non-numerical values (i.e., "Not a Number", NaN) were written in the input time series of the percentage of the population having access to safe water supply. This bug was detected after the calibration of the model variants and fixed in the runs.

The sectoral water use estimates end in different years. For the years thereafter, the value of the last data year was copied. The thermal electricity estimates end 2017 and manufacturing estimates 2016, whereas livestock estimates end already in 2011 (no change as compared to WaterGAP v2.2d except that the year 2011 was correctly used for prolonging the time series instead the year 2010 as done by accident in v2.2d) and domestic water use in 2010 (no temporal extension but the bugfix applied as described above).

### 2.3.1 Thermal electricity water use

WaterGAP estimates the amount of cooling water for thermal electricity production, both water abstractions and consumptive use, for each power plant individually. The input data for the location and capacity of thermal power plants is obtained from the World Electric Power Plants Data UDI (2020, last updated in 2010), along with relevant literature and case studies.

A thermoelectric power plant is defined as a power-generating facility that uses heat to generate energy, which may be produced by burning fossil fuels, biomass, or nuclear energy. Additionally, geothermal power plants and Concentrated Solar Power (CSP) plants, as well as other solar-related power plants that require water for cooling and cleaning of solar panels, have been incorporated into the database (Terrapon-Pfaff et al., 2020). Power plants that employ seawater or brackish water for cooling purposes are excluded. The time series of data on annual electricity production for different fuel types (EIA, 2021) as well as the thermal electricity water use time series was extended until the year 2017. The updated thermal electricity water use model was validated for the year 2015.

### 2.3.2 Manufacturing water use

The WaterGAP manufacturing water use model calculates the amount of water abstracted and consumed for production and cooling purposes in the manufacturing sector. A detailed model description can be found in Flörke et al. (2013) and Müller Schmied et al. (2021). The water use time series was prolonged to 2016 based on the key driving force manufacturing value added from Worldbank (2021).

## 2.4 New calibration dataset

The dataset of streamflow calibration stations was updated for WaterGAP v2.2e, now comprising a total of 1509 stations as compared to 1319 stations for WaterGAP v2.2d (Müller Schmied et al., 2021). An update was warranted as databases of

streamflow observations had been updated or newly established since the last station update roughly a decade ago and climate forcings now cover more recent years, e.g., until 2019 (Cucchi et al., 2020; Lange et al., 2021). As recent high-quality climate forcings are available only from 1979 onwards and require a concatenation to other less reliable climate forcings, with potential offsets (Müller Schmied et al., 2016), the update of the calibration stations also aimed at increasing the number of streamflow observations after 1978. A detailed description of the updating process can be found in Schiebener (2023).

### 2.4.1 Databases

As in the case of previous WaterGAP versions, the Global Runoff Data Center (GRDC) provides the main resource of stream-flow gauging station data. The GRDC database includes mostly daily streamflow time series of national data providers, but not all nationally available streamflow data are included. During the last few years, additional databases of streamflow indices have been made available.

The Global Streamflow Indices and Metadata Archive (GSIM) (Do et al., 2018; Gudmundsson et al., 2018) provides indices such as monthly streamflow for 30,000 stations from national daily streamflow data that have been collected, homogenized and enriched by metadata information. The start year of GSIM data is 1958.

The African Database of Hydrometric Indices (ADHI) (Tramblay et al., 2021) provides indices including monthly stream-flow for 1466 stations over the African continent, together with metadata. The start (end) year of ADHI data is 1950 (2018). While the GRDC database is continuously updated, this is not the case for GSIM and ADHI.

### 2.4.2 Station selection methodology

The criteria for considering a streamflow station to be suitable for the calibration of WaterGAP remain unchanged from Water-GAP v2.2d (Müller Schmied et al., 2014):

- an upstream area of at least 9,000 $\text{km}^2$

- a time series of at least four complete but not necessarily consecutive calendar years (with a maximum of two missing days per month)

- an inter-station catchment area of at least 30,000 $\text{km}^2$

The 1319 GRDC stations used for calibrating earlier model versions were identified in the GRDC metadata catalogue that was downloaded on July 30, 2021. Including updated streamflow data for these stations was straightforward as the location on the drainage network and criteria such as inter-station area had already been checked previously. Only one of the 1319 stations was no longer available in the GRDC database. For 175 stations, a change in the GRDC ID was considered. 119 additional GRDC stations that meet the criteria listed above and have a time series end after 1982 (to allow at least 4 years starting in 1979) were identified as potential additional stations. In total, 1437 stations with monthly data were downloaded from GRDC on August 6, 2021. Out of these, 1424 stations have four complete calendar years of data and are included in the new calibration data set of WaterGAP. 1565 GSIM and 197 ADHI stations that meet the spatial selection criteria were initially considered. Out

of these, 1367 GSIM stations and 189 ADHI meet the criterion of having four complete years of data and were included in the WaterGAP calibration data set.

The selected stations of all three data sources were plotted on the WaterGAP drainage network in order to 1) find and eliminate duplicates, which are not necessarily identified from the station metadata, 2) identify the stations that meet the interstation catchment area criteria and 3) re-map the station to a grid cell that fits with the drainage network. Re-mapping of the position focused on accurately relating the station either to the mainstream of the river or the tributary. A correcting factor for mismatches of drainage areas between the values provided by the station data producers and those calculated from the drainage direction map was not implemented but both areas can be found in the shapefiles of Müller Schmied and Schiebener (2022). As only GRDC is regularly updated, this data source was preferred in case of multiple stations with similar time series lengths in close-by grid cells. Time series of multiple stations in one grid cell were compared to further eliminate duplicates or to select the best-suited station. Where it was meaningful, time series were merged (e.g., for those cases where GSIM provides more recent years, but GRDC years before 1958). Furthermore, each time series was visually inspected in order to check the plausibility of data and to delete data points in case of obvious errors.

### 2.4.3  Resulting calibration dataset of streamflow observation

The final WaterGAP calibration dataset with streamflow observations consists of 1509 JSON files with monthly streamflow observations (only for years with values for all calendar months). Data for 1252 gauging stations originated from GRDC, 80 from ADHI and 177 from GSIM databases.

In WaterGAP calibration, 30 complete years of streamflow data are ideally used for model calibration. 949 of the 1509 stations have more than 30 years of data which requires a selection of a suitable start year for calibration. The later the global calibration start year is, the fewer stations and number of years are available for calibration (Fig. 2). In the case of 1979 as the start year for calibration, which would allow to use only the most reliable climate forcing, only 1375 out of 1509 gauging stations would be available for calibration. In addition, the number of years that would be available for calibration is reduced drastically in several parts globally (Fig. 3). Therefore, we decided to not constrain calibration to periods starting in 1979 or later.

The preferred period for calibration was set to 1981-2010. If for any gauging station, observation data are incomplete for this period, the following is done iteratively until 30 years of data are reached (not necessarily consecutive years) or until no further years are available for the station:

1. go back to 1979 as start year

2. extent the years after 2010

3. go back year by year starting from 1978 until 1901 as start year

During this counting procedure, the years 1980 and 1979 were accidentally considered twice. This led to the effect that for several stations, only 28 (for 362 stations) or 29 (for 34 stations) out of 30 possible calibration years are considered within the

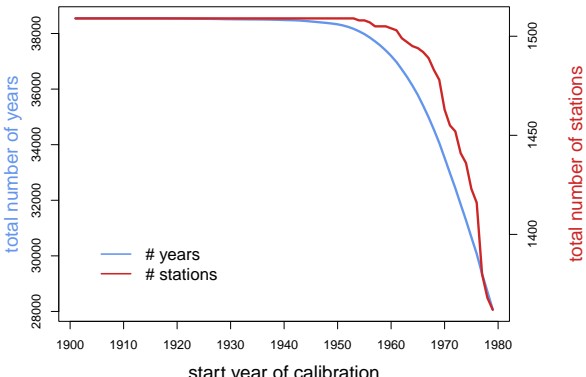

**Figure 2.** The number of gauging stations and years for calibration as a function of the year where the calibration starts. Both numbers decrease with a later start year of calibration indicating that the year 1916 is the most recent year to start the calibration without losing data points according to the station/data selection criteria. Note that the y-axes do not start at zero.

calibration procedure. Those missed years are always before 1978 and in the beginning of the possible calibration time period. An assessment of the difference of the correct 30 year time period and the erroneous one showed that for the majority of river basins, the difference of mean monthly streamflow is $< 5\%$ (Fig. S1). Due to this relatively small influence and as this issue was detected after all analyses had been conducted, we decided not to redo the calibration and all subsequent assessments.

In total, 38543 full calendar years could be used for calibrating WaterGAP v2.2e but due to the error described above, only 37785 full calendar years were considered. For a total of 993 (597 due to the error) out of 1509 stations, a 30-year period was available. For 336 of these stations, the 30-year period matches the time span 1981-2010. For 854 (825 due to the error) stations the calibration years (not necessarily 30 years) start before 1979 and out of these, 82 stations have all their calibration years before 1979. In contrast, the 1319 WaterGAP v2.2d calibration stations sum up to 31184 years, hence the update of the

calibration dataset increased the number of years by around 24% (21% due to the error). In terms of calibration area, the overall process increased the calibration area by $2.14\,10^6\mathrm{km}^2$, whereas $0.53\,10^6\mathrm{km}^2$ are no longer included in the calibration area e.g., due to suspicious data (Fig. 4). This results in an increase of calibrated drainage area from 53.8% in WaterGAP v2.2d to 55.1% in WaterGAP v2.2e of the global land area outside Antarctica and Greenland. The average basin size (excluding any additional upstream basin area) decreased from $54{,}000\ \mathrm{km}^2$ in v2.2d to $48{,}300\ \mathrm{km}^2$ in v2.2e. The calibration basins and streamflow time

series are provided in Müller Schmied and Schiebener (2022).

## 2.5   New handling of inland sinks

Cells that represent inland sinks, i.e., cells without outflow of liquid water, are handled like any other cell in WaterGAP v2.2d. Since WaterGAP v2.2a (Döll et al., 2014), focused groundwater recharge below surface water bodies (i.e., lakes and wetlands) is calculated in (semi)arid grid cells. In the case of (semi)arid inland sinks, the focused recharge can reach very high values,

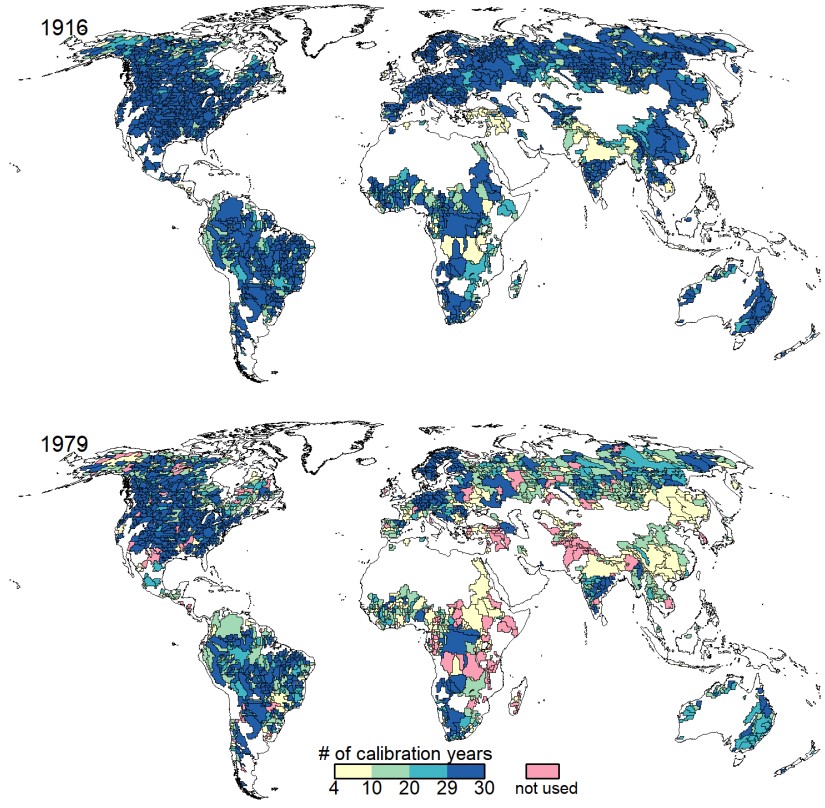

**Figure 3.** Number of complete years usable for calibration of model parameters in the calibration basins shown, for 1916 and 1979 as calibration start years. "not used" refers to the case where less than four years of streamflow data are available the case of starting calibration in 1979 such that these basins would not be included in model calibration.

which limits assessment of this variable, e.g., in climate impact studies. Furthermore, it is unrealistic to provide a streamflow value for an inland sink as there is - other than an ocean outflow cell - no grid cell that could receive the streamflow generated in inland sinks.

Hence, inland sinks are handled in v2.2e as follows:

 – no focused groundwater recharge below surface water bodies,

– surface runoff and groundwater outflow are routed to the surface water bodies (no fractional routing, Döll et al. (2014))

 – simulated streamflow of inland sinks is added to actual evapotranspiration in model output and streamflow is set to zero.

This new handling leads to correctly calculated renewable water resources in inland sinks, which can become negative as all precipitation and cell inflow is assumed to be evapotranspired. Diffuse groundwater recharge is computed, and groundwater abstractions as well as surface water abstractions from lakes are taken into account in modeling inland sinks. As a consequence

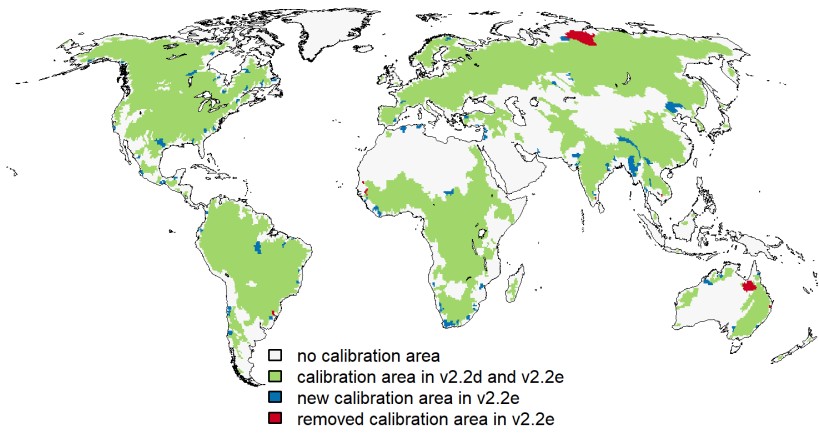

**Figure 4.** Areas considered for calibration in WaterGAP versions v2.2d and v2.2e. Blue colors indicate grid cells that are newly present as calibration area in v2.2e due to the update of the data basis whereas red colours show grid cells that are no longer calibrated in v2.2e in comparison to v2.2d.

of setting streamflow to zero in inland sinks, the reservoir algorithm cannot be initialized in those grid cells and thus in total four global reservoirs in inland sink cells are treated as global lakes in WaterGAP v2.2e.

## 3  New options for special model applications

### 3.1  Alternative PET calculation method to approximate the effect of vegetation response when estimating the impact of climate change on evapotranspiration

Potential evapotranspiration on land surfaces (PET) is determined by a combination of plant transpiration and evaporation from the canopy and the soil. As such, PET is influenced by vegetation characteristics and processes that are affected by human-induced climate change, in particular rising atmospheric $CO_2$ concentrations. The physiological effect (with closing stomata decreasing transpiration), the structural effect (also known as the fertilization effect, which may increase canopy evaporation and transpiration), and biome shifts are three types of vegetation responses to rising atmospheric $CO_2$ (Gerten

et al., 2014). These effects influence PET, and if not accounted for, lead to wrong estimates of the impact of climate change on evapotranspiration and water resources.

Typical hydrological models, such as WaterGAP, do not simulate the plant phenology processes leading to these effects nor the interaction with the atmosphere. This significantly constrains the capacity of standard hydrological models to assess how water resources change under climate change. Given the intricacy and considerable uncertainty associated with simulating

vegetation responses, Peiris and Döll (2023) recommended running hydrological models in two variants, one with the PET algorithm used for conditions where PET is not impacted by vegetation response to climate change (i.e., the standard PET) and

the other where this impact is approximated. Accordingly, in WaterGAP v2.2e, the Priestley-Taylor (PT) method is used in the standard model runs to calculate PET (Müller Schmied et al., 2021) and the Priestley-Taylor modified approach (PT-MA) is applied as the alternative PET computation method, where PT-MA considers the vegetation effect when computing the PET in 300 a very simple and approximate way.

The PT method computes PET as a function of net radiation and temperature, where PET increases with temperature. However, analyzing evaporation changes of an ensemble of global climate models, Milly and Dunne (2016) found that under future climate change, PET change as computed with the PT method overestimates the increase of future PET and the PET change is a function of net radiation change only. The impact of increasing temperature on PET is approximately canceled by 305 the impact of changes of other processes that are taken into account by GCMs but not by typical hydrological models (Milly and Dunne, 2016; Yang et al., 2019).

The new PET method PT-MA, which was developed based on the results of Milly and Dunne (2016), can be applied for estimating hydrological changes due to climate change between a reference period and a future period. A temperature reduction factor $T_{diff}$ is calculated in pre-processing for each land grid cell and year in the future time period and stands for the difference 310 between the annual mean temperature of a 20-year period centered around the year of interest and the mean annual temperature of the reference period. The model then applies this temperature reduction factor to adjust the daily temperature values in future scenarios, thus removing the long-term temperature trends. As a result, the model computes future PET by only taking into account changes in net radiation only, while still varying temperatures at daily to inter-annual scales.

The PT-MA method leads to a roughly similar effect of future anthropogenic climate change on PET as computed by the 315 ensemble of GCMs. Therefore, the PET-MA method is applicable as an alternative for estimating the change of hydrological variables between the reference period and a period in the future for the case, as different from standard WaterGAP, it does not neglect the impact of vegetation dynamics on actual evapotranspiration and thus runoff. With decreased evaporation as compared to climate change runs with the standard WaterGAP with PT, the PT-MA runs lead to less drying or more wetting than PT runs. Given the very simplified manner of considering the vegetation response to climate change, we recommend using 320 both the PT and PT-MA model variants in an ensemble approach to estimating hydrological hazards of climate change. Peiris and Döll (2023) provides further details and verification of this approach.

## 3.2 Integration of glaciers

WaterGAP v2.2d neither simulates water storage in glaciers nor water flows related to glacier dynamics. To take into account water storage and flow dynamics of glaciers in WaterGAP, we implemented a glacier algorithm in WaterGAP v2.2e. This 325 algorithm reads input data sets of glacier area and glacier mass change computed with the global glacier model of Marzeion et al. (2012) and of total precipitation (rainfall plus snowfall) on glacier area from the atmospheric data set used to force the glacier model. These input data sets are used 1) to integrate a glacier area fraction in the grid cells where glaciers are located, 2) to calculate glacier runoff, i.e., the runoff generated from precipitation on glacier area and glacier mass change, and 3) to include a glacier water storage compartment in the hydrology model. The glacier runoff is added to the cell's fast runoff, 330 which partly flows directly into the river while the rest flows into the other surface water bodies. In the standard version of

WaterGAP v2.2e, the glacier algorithm is switched off, i.e., glaciers are not included. This is because the algorithm relies on glacier-related input data sets that are currently only available from January 1948 to December 2016, whereas standard model runs require input data from 1901 onwards and up-to-date climate forcing datasets prolongs after the year 2016. WaterGAP v2.2e with glaciers was validated by comparing simulated global monthly terrestrial water storage anomalies to observations from an ensemble of four GRACE spherical harmonic solutions for the period January 2003 to August 2016. For more details regarding the glacier algorithm implementation and validation, we refer the reader to Cáceres et al. (2020).

### 3.3 Calculation of river water temperature

The estimation of water temperature of rivers is relevant e.g., for the solubility of gases, the metabolic rate of aquatic flora and fauna and the formation of ice. Furthermore, changes in water temperature do not only have local but also downstream effects (Olden and Naiman, 2010). Also, the return flows from thermal power plants increase river water temperature. Due to the importance of water temperature as a physical water quality indicator, the Inter-Sectoral Impact Model Intercomparison Project (ISIMIP) included river water temperature as a requested variable in its recent project phase 3. In WaterGAP v2.2e, and inspired by the approaches of Van Beek et al. (2012) and Wanders et al. (2019), the calculation of river water temperature is implemented. Implementation details as well as a validation against observed river water temperature can be found in Ackermann (2023). When comparing simulated river temperatures of WaterGAP with a regression approach of air temperature (Punzet et al., 2012) results are rather similar. Ackermann (2023) initially compared the results of WaterGAP and the regression approach with observation data and concluded that the regression approach from air temperature obtains often higher performance indicator values. They also showed that e.g., the inclusion of warming due to return flows from thermal power plants improved model simulations. For assessing if the implemented approach is useful for impact assessments, further evaluation is required and will be conducted e.g., in the newly formed water quality sector of ISIMIP.

### 3.4 Ability to start from prescribed initial conditions

A typical model run of WaterGAP starts with several years of initialization (e.g., 5 years) to enable storage compartments to swing in from their initial conditions to more realistic ones. The stop and restart of the model in a specific month was a functionality that was not required in earlier versions of WaterGAP. WaterGAP v2.2e is now able to store all states (storage compartments), parameters (such as area reduction factors) and additional information (such as days of the vegetation growing period) for a pre-defined month of a specific year. A model run can then be started from this prescribed stored initial state.

The ability to start the model from a prescribed initial condition is required, for example, for model runs for near-real-time monitoring and ensemble forecasts. This feature was used within the framework of the ISIMIP3b simulations as different scenarios for the future time period could be started from a given state of the historical time period, which reduced runtimes as compared to a transient run drastically.

Furthermore, this functionality enables the model to run a certain month, modify e.g., storage compartments externally (assimilation of e.g., GRACE data) and start the next month in WaterGAP. This offline-coupling allows data assimilation studies and in addition, WaterGAP is prepared for online-coupling in the PDAF system (Nerger and Hiller, 2013). For this,

**Table 1.** Overview of the climate forcings used to drive WaterGAP v2.2e (and v2.2d).

| no | name | before 1979 | after 1979 | temporal coverage | source and further info |
|---|---|---|---|---|---|
| 1 | gswp3-w5e5 | GSWP3 v1.09 | W5E5 v2.0 | 1901-2019 | Lange et al. (2022) |
| 2 | gswp3-era5 | GSWP3 v1.09 | ERA5 | 1901-2022 | provided by Stefan Lange[1] |
| 3 | 20crv3-w5e5 | 20CRv3 | W5E5 v2.0 | 1901-2019 | Lange et al. (2022) |
| 4 | 20crv3-era5 | 20CRv3 | ERA5 | 1901-2021 | Lange et al. (2022) |

[1] until 2021, extended to 2022 by the authors of this paper based on the methodology provided by Stefan Lange

WaterGAP compiles not only as an executable to run on a Linux system but also as a library that can be embedded in PDAF. As the writing and reading of physical data is omitted, this online coupling strongly reduces the runtime of monthly data assimilation.

## 4    Climate forcings and model setup

### 4.1    Climate forcings

WaterGAP was calibrated and run with a total of four climate forcings, which are mainly from the ISIMIP phase 3a (Frieler et al., 2023). All the climate forcings are a concatenation of two data sets - one for the period prior to 1979 and one for the period starting in 1979 (Table 1). The year 1979 is the first year of the current up-to-date ERA5 reanalysis, which is either directly used or the basis for a specific bias adjustment to observation data.

GSWP3 in its version 1.09 (Kim, 2017) is a bias-adjusted and downscaled version of the Twentieth Century Renalysis version 2 (20CRv2) (Compo et al., 2011). The ensemble member 1 of the Twentieth Century Reanalysis version 3 (20CRv3) (Slivinski et al., 2019, 2021) was interpolated to 0.5 deg spatial resolution but not bias-adjusted (Lange et al., 2022). ERA5 (Hersbach et al., 2020) is the latest version of the European Reanalysis. The year 2022 for ERA5 is added based on the scripts that have been provided by Stefan Lange with an ERA5-download date of 25.01.2023. W5E5 v2.0 (Cucchi et al., 2020; Lange et al., 2021) is a bias-adjusted version of the current version of the European Reanalysis ERA5 (Hersbach et al., 2020).

The climate forcings are concatenated by applying a bias adjustment of the dataset before 1979 to the dataset thereafter by using ISIMIP3BASD v2.5.1 (Lange, 2019, 2021). This reduces discontinuities at the 1978/1979 transition. For details see Mengel et al. (2021).

### 4.2    WaterGAP model variants

The standard model variant "ant" includes human interference with the hydrological cycle, namely human water use and reservoir operation (in ISIMIP3 nomenclature "histsoc"). In contrast, the model is also run in a "nat" mode without water use and reservoirs to reflect a hydrological system without those direct human impacts (in ISIMIP3 nomenclature "nowatermgt"). All model variants are calibrated with the corresponding climate forcing. The standard climate forcing of WaterGAP v2.2e

**Table 2.** Global water balance components with a model variant of WaterGAP v2.2e including local reservoirs in local lakes under naturalized variant (as it was in v2.2d, labeled v2.2e_nat with local reservoirs) and in WaterGAP v2.2e where local reservoirs are removed from local lakes in a naturalized variant (labeled v2.2e_nat). Water balance components for the time period 1991-2019. All units in $km^3\,yr^{-1}$.

|  | v2.2_nat with local reservoirs | v2.2e_nat | v2.2e - v2.2e with local reservoirs |
|---|---|---|---|
| Precipitation | 111578.0 | 111578.0 | 0.0 |
| Actual evapotranspiration | 70863.7 | 70852.5 | -11.3 |
| Streamflow into oceans | 40709.4 | 40720.7 | 11.3 |
| Change of total water storage | 4.8 | 4.8 | 0.0 |
| Long-term average volume balance error | 0.0 | 0.0 | 0.0 |

is gwsp3-w5e5. To compare the effect of model development, we calibrated and ran WaterGAP v2.2d with the gswp3-w5e5 climate forcing and the calibration database of v2.2e. In total, the outputs of eight WaterGAP v2.2e variants are available (four climate forcings with "ant" and "nat" setup) as well as the output of two WaterGAP v2.2d variants (one climate forcing with "ant" and "nat" setup, calibrated to the new WaterGAP v2.2e streamflow observations data).

## 5 Results of standard model modifications

### 5.1 Effect of removing local reservoirs from naturalized runs

The impact on the global water balance of no longer assuming that local reservoirs exist in naturalized runs is small (Table 2). As fewer water bodies are considered in v2.2e, actual evaporation decreases and streamflow increases by the same amount. Streamflow thus increases by less than 0.03 %. The change in water storage components is only minor (not shown).

### 5.2 New calibrated parameters

The calibration as implemented in the standard version of WaterGAP focuses on adjusting biases in a rather simple method. More comprehensive approaches are currently in development (Döll et al., 2024; Hasan et al., 2023) might be used in future model versions. While the calibration approach for WaterGAP v2.2e is the same as for WaterGAP v2.2d, the data set of observed streamflow differs as described in Sect. 2.4. Calibration of WaterGAP v2.2e was done for all four climate forcings. To explore the impact of the model version, WaterGAP v2.2d driven by gswp3-w5e5 was calibrated using the v2.2e streamflow observation dataset, too. As described in Müller Schmied et al. (2021, their Section 4.9), the calibration follows a four-step scheme with specific calibration status (CS):

1. CS1: adjust the basin-wide uniform parameter $\gamma$ (Müller Schmied et al., 2021, their Eq. 18) in the range of [0.1-5.0] to match mean annual observed streamflow within $\pm$ 1%.

2. CS2: adjust $\gamma$ as for CS1, but within 10% uncertainty range (90-110% of observations).

3. CS3: as CS2 but apply the areal correction factor CFA (adjusts runoff and, to conserve the mass balance, actual evapo-transpiration as the counterpart of each grid cell within the range of [0.5-1.5]) to match mean annual observed streamflow with 10% uncertainty.

4. CS4: as CS3 but apply the station correction factor CFS (multiplies streamflow in the cell where the gauging station is located by an unconstrained factor) to match mean annual observed streamflow with 10% uncertainty to avoid error propagation to the downstream basin.

For each basin, calibration steps 2-4 are only performed if the previous step was not successful.

The calibration of WaterGAP v2.2e (v2.2d) driven by the standard climate forcing gwsp3-w5e5 results in 519 (524) basins with calibration status CS1, 216 (212) basins with calibration status CS2, 262 (323) basins with calibration status CS3 and 512 (449) basins with calibration status CS4. While, with 49 %, the percentage of river basins that can be calibrated without applying correction factors is nearly the same for both model versions, the modification/update of reservoir or water use data in v2.2e lead to substantially more stations where not only the areal correction factor CFA but also the station correction factor CFS is required to match the simulated long-term annual streamflow with observations. The 69 stations that moved from CS3 in WaterGAP v2.2d to CS4 in WaterGAP v2.2e are located all around the globe in different climate zones, but a lot of them are located in snow-dominated regions. 64 of these stations have a CFS value of larger than 1 indicating streamflow is underestimated by WaterGAP v2.2e unless CFS is applied. This difference is due to a slightly different handling of the calibration routines of v2.2d and v2.2e. Whereas in v2.2d the calibration period uses a spin-up of a 5 year time period prior the calibration start year, in v2.2e the calibration start year is repeated 5 times. Hence, different calibration results can occur especially in the first calibration year which can finally result in a different CS.

The spatial distribution of calibration parameters and the calibration status is shown for WaterGAP v2.2e and the standard forcing gwsp3-w5e5 in Fig. 5, and for v2.2d in supplementary Figure S2. For the calibration results for WaterGAP v2.2e driven by the other three climate forcings, the reader is referred to the supplementary Figs. S3-S5.

## 5.3 Improved handling of inland sinks

The improved handling of inland sinks leads to a reduction of global streamflow, an increase in actual evapotranspiration and a slight decrease of the total water storage change in the period 2001-2010 (Table 3). This is expected as streamflow is now assumed to become actual evapotranspiration in inland sinks. Hence, between WaterGAP v2.2d and WaterGAP v2.2e, the assessment of streamflow into oceans in the water balance component has a different meaning. The improved handling of inland sinks increases global actual evapotranspiration by 1.1 % and decreases global streamflow into oceans and inland sinks by 2.0 %. Focused recharge is neglected in inland sinks which leads to less groundwater storage. The water balance error is not affected.

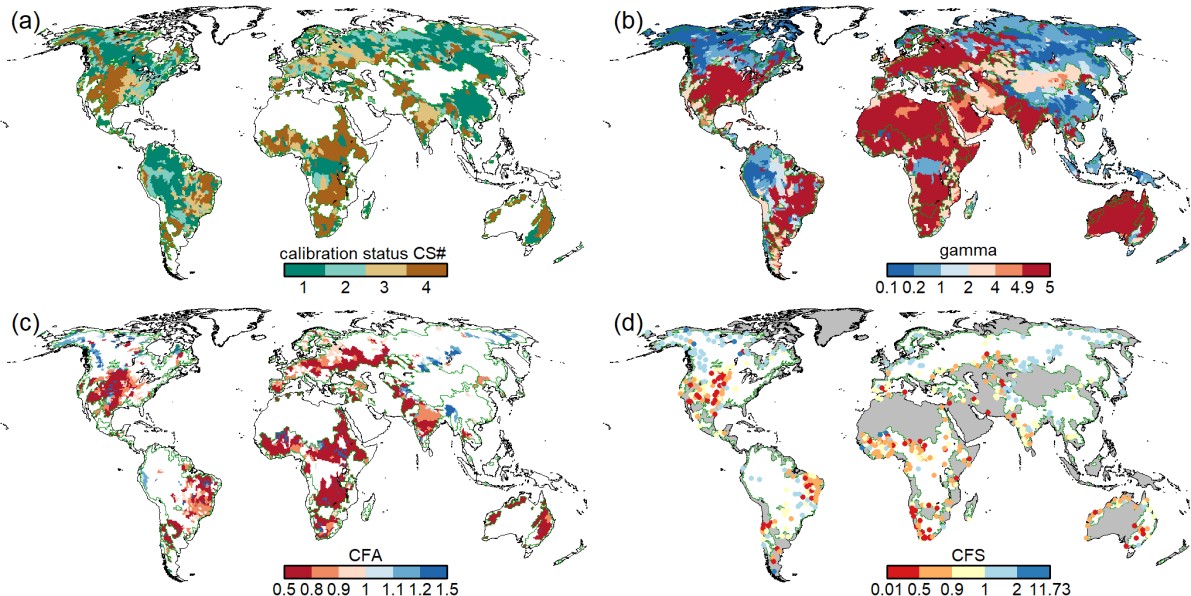

**Figure 5.** Results of the calibration of WaterGAP v2.2e driven by the gswp3-w5e5 climate forcing, with (a) the calibration status of each of the 1509 calibration basins, (b) calibration parameter $\gamma$, (c) areal correction factor CFA and (d) station correction factor CFS. Grey areas in (d) indicate regions with regionalized calibration parameter $\gamma$ and for (a)-(d) dark green outlines indicate the boundaries of the calibration basins. For details of the calibration procedure, the reader is referred to Müller Schmied et al. (2021)

.

**Table 3.** Global water balance components with a model version including the improved handling of inland sinks in WaterGAP v2.2e as compared to previous handling (as in WaterGAP v2.2d). Water balance components for the time period 2001-2010. Please note that the model version used for this assessment is a pre-v2.2e version and run with a different climate (a combination of WFD-WFDEI). The purpose here is only to show the effect of new handling of inland sink cells. The unit of all variables is $\text{km}^3 \text{ yr}^{-1}$.

|  | v2.2e old inland | v2.2e standard | v2.2e st - v2.2e old |
| --- | --- | --- | --- |
| Precipitation | 112438.5 | 112438.5 | 0.0 |
| Actual evapotranspiration[1] | 72086.8 | 72903.8 | 817.0 |
| Streamflow into oceans[2] | 40332.4 | 39518.6 | -813.8 |
| Change of total water storage | 19.3 | 16.0 | -3.3 |
| Long-term average volume balance error | 0.1 | 0.1 | 0.0 |

[1] including (excluding) streamflow in inland sinks for v2.2e (v2.2d)

[2] including (excluding) streamflow in inland sinks for v2.2d (v2.2e)

## 5.4 Global water balance components

### 5.4.1 Major water balance components

The calculation of globally aggregated water balance components for WaterGAP v2.2e driven by gswp3-w5e5 is shown in
Table 4. The corresponding tables for the other model variants are provided in the supplementary Tables S1-S4. Due to bias
adjustment of precipitation, precipitation is larger for the climate forcings that include W5E5 as compared to those that include
ERA5. For all model variants, climate forcings and time periods, the streamflow to the oceans (for Table S1 it is streamflow
to the oceans and inland sinks) is between 39,000 and 40,500 $km^3yr^{-1}$. As global streamflow does not vary much as a
consequence of calibration even though the precipitation varies, actual evapotranspiration differs strongly between the model
variants that are driven by either W5E5 or ERA5, from 70,000 to 80,000 $km^3yr^{-1}$. Please note that as a consequence of the
new handling of inland sinks (Sect. 2.5), inland sinks do not contribute to globally aggregated streamflow in WaterGAP v2.2e
and thus the amount is lower than in previous model versions. However, we indicated the inflow into inland sinks in the tables
for model version v2.2e, which is the amount of water that would have been included in row 3 for model version v2.2d but is
now included in row 2. For the Table S1 (WaterGAP v2.2d) row 4 is included in row 3. This different handling of inland sinks
explains the differences between streamflow and actual evapotranspiration between versions v2.2d and v2.2e. For assessments
of renewable water resources, it is recommended to sum up rows 3 and 4 for WaterGAP v2.2e results.

**Table 4.** Global-scale (excluding Antarctica and Greenland) water balance components for different time spans as simulated with WaterGAP
v2.2e with gswp3-w5e5. The unit of all variables is $km^3yr^{-1}$. Long-term average volume balance error is calculated as the difference of
component 1 and the sum of components 2,3 and 8.

| No. | Component | 1961 – 1990 | 1971 – 2000 | 1981 – 2010 | 1991 – 2019 | 2001 – 2019 |
|-----|-----------|-------------|-------------|-------------|-------------|-------------|
| 1 | Precipitation | 110637 | 111279 | 111350 | 111574 | 111655 |
| 2 | Actual evapotranspiration[1] | 71325 | 71755 | 71816 | 71998 | 72063 |
| 3 | Streamflow into oceans | 39295 | 39530 | 39584 | 39666 | 39697 |
| 4 | Inflow into inland sinks[2] | 776 | 794 | 795 | 841 | 846 |
| 5 | Actual consumptive water use[3] | 904 | 1049 | 1195 | 1307 | 1369 |
| 6 | Actual net abstraction from surface water | 1036 | 1186 | 1338 | 1448 | 1501 |
| 7 | Actual net abstraction from groundwater | -132 | -137 | -143 | -141 | -132 |
| 8 | Change of total water storage | 17 | -6 | -49 | -91 | -105 |
| 9 | Long-term average volume balance error | -0.46 | -0.34 | -0.20 | -0.08 | -0.07 |

[1] including actual consumptive water use

[2] streamflow that flows into inland sinks; the simulated streamflow of inland sinks is added to actual evapotranspiration

[3] sum of rows 6 and 7

### 5.4.2 Water storage components

The globally aggregated water storage component changes are shown in Table 5 for WaterGAP v2.2e driven by gswp3-w5e5. While during the period 1961-1990, the increase in water storage in reservoirs and regulated lakes, due to dam construction, more than balance the decrease of groundwater storage due to human water use, the latter dominated in all later evaluation periods. While the annual rate of groundwater loss has steadily increased from the period 1961-2000 to the period 2001-2019, the annual total water storage loss rate has steadily increased from the period 1971-2000 onward. This is also true for the other model variants (supplementary Tables S6-S9). For all three climate forcings, WaterGAP v2.2e computes a decline in snow water storage since the period 1981-2010. For other storage compartments, different climate inputs result in different signs of change, without a specific component that is dominantly sensitive. When comparing the water storage changes of WaterGAP v2.2e (Table 5) and WaterGAP v2.2d (Table S5), most components are similar but in WaterGAP v2.2d the reservoirs and global lakes gain less water than in WaterGAP v2.2e in the more recent time periods.

**Table 5.** Globally aggregated (excluding Antarctica and Greenland) water storage component changes during different periods as simulated by WaterGAP v2.2e with gswp3-w5e5. All units in $km^3 yr^{-1}$.

| No. | Component | 1961–1990 | 1971–2000 | 1981–2010 | 1991–2019 | 2001–2019 |
|-----|-----------|-----------|-----------|-----------|-----------|-----------|
| 1 | Canopy | 0 | 0 | 0.1 | 0 | 0 |
| 2 | Snow | 11.4 | -9.2 | -2.5 | -13.7 | -0.8 |
| 3 | Soil | 4.9 | 7.6 | 9.5 | -0.3 | -8.8 |
| 4 | Groundwater | -62.0 | -68.4 | -96.0 | -117.7 | -144.5 |
| 5 | Local lakes | 0.3 | 1.1 | 0.9 | 0.2 | -1.3 |
| 6 | Local wetlands | 0.7 | -0.5 | 4.6 | 4.4 | 9.2 |
| 7 | Global lakes | -2.7 | -3.5 | -2.5 | 4.3 | 9.8 |
| 8 | Global wetlands | -3.5 | 5.0 | 0.8 | 0.0 | -7.0 |
| 9 | Reservoirs and regulated lakes | 70.8 | 50.8 | 36.0 | 24.9 | 25.1 |
| 10 | River | 0.4 | 5.4 | -8.1 | 3.8 | 4.1 |
| 11 | Total water storage | 20.3 | -11.9 | -57.2 | -94.1 | -114.3 |

### 5.4.3 Water use components

Globally aggregated sectoral potential withdrawal and consumptive water uses as well as use fractions from groundwater are shown in Table 6 for WaterGAP v2.2e and gswp3-w5e5; the corresponding values for the other model variants are given in the supplementary Tables S10-S13. Irrigation accounts for two thirds of potential water abstractions (WU) and 88% of potential consumptive use. Groundwater withdrawals are estimated to cover about 22% of all withdrawals, with the highest fraction for the domestic sector, while 35% of total potential consumptive use is supplied by groundwater, due to the assumed higher water use efficiency in the case of irrigation with groundwater. The table values represent the human demand for water that cannot be

**Table 6.** Globally aggregated (excluding Antarctica and Greenland) sectoral potential withdrawal water use WU and consumptive water use CU ($km^3yr^{-1}$) as well as use fractions from groundwater (%) as simulated by GWSWUSE of WaterGAP v2.2e for the time period 1991-2019.

| Water use sector | WU | Percent of WU from groundwater | CU | Percent of CU from groundwater |
|---|---|---|---|---|
| Irrigation | 2541 | 25 | 1179 | 37 |
| Thermal power plants | 592 | 0 | 18 | 0 |
| Domestic | 352 | 35 | 57 | 36 |
| Manufacturing | 298 | 27 | 60 | 25 |
| Livestock | 29 | 0 | 29 | 0 |
| Total | 3813 | 22 | 1342 | 35 |

completely satisfied in WaterGAP v2.2e due to a lack of surface water resources. Only 1307 $km^3yr^{-1}$ of the 1342 $km^3yr^{-1}$ of potential consumptive use can be fulfilled in the period 1991-2019 (row 5 in Table 4). The climate forcings including ERA5 have 150 $km^3yr^{-1}$ less potential withdrawal water use for irrigation than the forcings with W5E5, which is a result of more precipitation and thus less irrigation demand. Still, potential consumptive use of 1268 $km^3yr^{-1}$ cannot be fulfilled, and only 1237 $km^3yr^{-1}$ is actually consumed (compare Tables S13 and S5). Global sectoral water demand differences between WaterGAP v2.2d (Table S9) and v2.2e are visible only for the two updated water use sectors cooling of thermal power plants and manufacturing.

## 6 Application of new model options

### 6.1 Effect of PET calculation with PT-MA on the global water balance under climate change

The effect of the modified Priestley-Taylor PET approach (PT-MA) is tested by running WaterGAP as driven by two ISIMIP3b global climate models (GFDL-ESM4 and CanESM5) for the future under the emissions scenario RCP8.5, with both standard PT and the newly developed PT-MA approach. Analyzing the global water balance components for the period of 2071-2100, actual evapotranspiration is, as expected, lower with the PT-MA method, and global streamflow is increased by around the same amount (Table 7). In the case of GFLD-ESM4 and CanESM5, the PT-MA method leads to an increase of the streamflow into oceans by 2.7% and 4.0%, respectively. If hydrological models neglect the effect of the active vegetation response to the increasing atmospheric $CO_2$ concentrations, it can thus be expected that they may underestimate future water resources (Milly and Dunne, 2016; Peiris and Döll, 2023). Other water balance components are affected only marginally, also because the PT-MA method is not applied in WaterGAP v2.2e when computing irrigation water use.

**Table 7.** Globally aggregated (excluding Antarctica and Greenland) water balance components for the period 2071-2100 as computed with standard PET model variant (PT) and the alternative PET model variant (PT-MA) that takes into account in a very simple manner the impact of climate change on vegetation when computing PET. The WaterGAP variants are driven by the bias-adjusted output of the GFDL-ESM4 and CanESM5 as provided by ISIMIP. The column diff corresponds to PT-MA - PT for the respective GCM. All units in $km^3 \, yr^{-1}$.

| | $GFDL_{PT}$ | $GFDL_{PT-MA}$ | diff | $CanESM5_{PT}$ | $CanESM5_{PT-MA}$ | diff |
|---|---|---|---|---|---|---|
| Precipitation | 108633 | 108633 | 0 | 130617 | 130617 | 0 |
| Actual evapotranspiration[1] | 70924 | 69907 | -1017 | 82838 | 80894 | -1944 |
| Streamflow into oceans[2] | 37850 | 38859 | 1009 | 47764 | 49689 | 1925 |
| Change of total water storage | -141 | -133 | 8 | 15 | 34 | 18 |
| Long-term average volume balance error | 0 | 0 | 0 | 0 | 0 | 0 |

[1] including actual consumptive water use

[2] inland sinks are not considered

## 6.2 Effect of glaciers on the global water balance

The inclusion of glaciers in a WaterGAP run influences all global water balance components (Table 8). Precipitation is higher due to a different precipitation product used in the original glacier model (see Cáceres et al., 2020), so that the other components are impacted by both the different precipitation and the glacier processes themselves. As expected, total water storage shows much stronger negative trends if the glacier option is enabled due to ice loss of the melting glaciers. Global streamflow into oceans increases with enabled glacier option due to 1) the additional meltwater from the glaciers, 2) increased precipitation input and 3) decreased AET, as AET is assumed to be zero on the areas that are covered by glaciers but is larger than zero when standard land cover takes up the part of the glacier in the standard run. Other components are affected only marginally. A comparison of simulated terrestrial water storage anomalies (TWSA) averaged over all land areas of the globe (except Antarctica and Greenland) to GRACE TWSA observations showed a good fit regarding seasonality and trend, while without the glacier options, the simulated WaterGAP trend is too small (Cáceres et al., 2020).

## 7 Evaluation of WaterGAP v2.2e

### 7.1 Model variants used for the evaluation

The evaluation was done using the output of the WaterGAP runs in the anthropogenic mode, considering human water use and reservoir operation. The difference between the model version v2.2d and v2.2e is investigated by running both variants with the climate forcing gswp3-w5e5. The effect of the different climate forcings is assessed by comparing WaterGAP v2.2e driven by the gswp3-w5e5 climate forcing to WaterGAP driven by the gswp3-era5 climate forcing. The evaluation as such follows closely Müller Schmied et al. (2021) for the sake of consistency.

**Table 8.** Global-scale (excluding Antarctica and Greenland) water balance components for two time spans as simulated with the standard model version WaterGAP v2.2e and the version with enabled glacier option. All units in $\mathrm{km^3\ yr^{-1}}$. Long-term average volume balance error is calculated as the difference between component 1 and the sum of components 2, 3 and 7.

| No. | Component | 1971 - 2000 | | | 2001-2016 | | |
|---|---|---|---|---|---|---|---|
| | | standard | glacier | glacier-standard | standard | glacier | glacier-standard |
| 1 | Precipitation | 111279 | 111955 | 676 | 111601 | 112254 | 653 |
| 2 | Actual evapotranspiration[1] | 71756 | 71642 | -114 | 72043 | 71930 | -112 |
| 3 | Streamflow into oceans and inland sinks | 39529 | 40438 | 909 | 39696 | 40735 | 1039 |
| 4 | Actual consumptive water use[2] | 1049 | 1057 | 8 | 1364 | 1371 | 7 |
| 5 | Actual net abstraction from surface water | 1186 | 1206 | 20 | 1492 | 1510 | 18 |
| 6 | Actual net abstraction from groundwater | -137 | -149 | -12 | -128 | -139 | -11 |
| 7 | Change of total water storage | -6 | -124 | -118 | -138 | -412 | -274 |
| 8 | Long-term average volume balance error | -0.34 | -0.34 | 0.00 | -0.09 | -0.09 | 0.00 |

[1] including actual consumptive water use

[2] sum of rows 5 and 6

## 7.2 Independent datasets used for model evaluation

### 7.2.1 Water abstractions

AQUASTAT is the Food and Agriculture Organization of the United Nations Global Information System on Water and Agriculture (FAO, 2022). For individual countries, it provides water abstractions (withdrawals) for different water use sectors. In addition to the six water use variables used in Müller Schmied et al. (2021), we used here abstractions for cooling of thermoelectric power plants as well as those for the livestock sector. For the evaluation, all database entries (yearly values) available on FAO (2022) until (including) 2019 were used. The evaluation metrics as described in Müller Schmied et al. (2021, their Sect. 6.3.1) are calculated using each single data point of AQUASTAT without any temporal aggregation by country.

### 7.2.2 Streamflow

The streamflow dataset described in Sect. 2.4 and Müller Schmied and Schiebener (2022) can be classified as follows.

- all months available for the station, including months in incomplete years (ALL)

- months in complete years that went into the calibration of the model (CAL)

- months that remain from ALL when months for CAL are removed (VAL)

The number of months per basin and class is shown in Fig. 6. Those basins (stations) that have less then 361 months in total and consequently for calibration, do not have additional streamflow data for validation. The median number of months

per category are 544, 336 and 207 for ALL, CAL and VAL, respectively. For VAL, 240 of the 1509 calibration basins have less than 12 months with observations (out of which 198 are without any observations). This means that for around 16 % of the basins, validation is not possible. For this reason, and also as model calibration only aims at improving long-term average annual streamflow, we evaluated the simulated monthly streamflow time series against all available monthly observations in the following but provide the same assessments with CAL and VAL in the supplementary material.

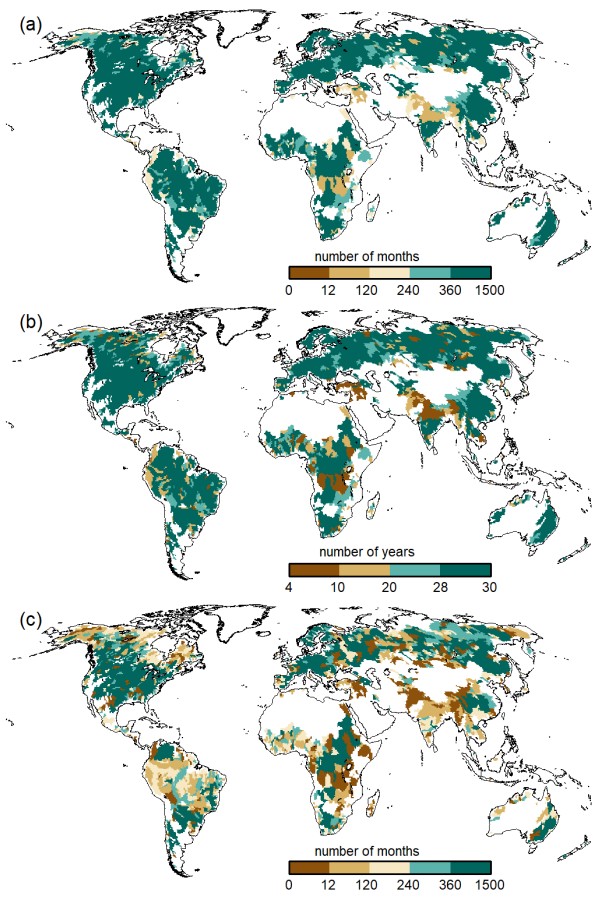

**Figure 6.** Number of available months of streamflow observation data (All) (a), number of complete years for calibration (CAL) (b) and number of months for validation (VAL) (c).

### 7.2.3 Terrestrial water storage anomalies

The Gravity Recovery And Climate Experiment (GRACE) satellite mission was in orbit between 2002 to 2017 to observe the temporal changes of the Earth's gravity field and obtain monthly time series of terrestrial water storage anomalies (TWSA). Its follow-on mission GRACE-FO started in 2018 to continue the measurements. Thus, a data gap of several months exists. In addition, due to the aging batteries of the GRACE mission, no data were collected in specific periods leading to further

data gaps in the GRACE time series. Forootan et al. (2020) published a strategy based on independent component analyses (ICA) to combine data from the Swarm explorer mission and GRACE(-FO) to reconstruct a gap-free time series. The AAU Geodesy product was recently extended to include GRACE-FO TWSA data until 07/2021. For the reconstruction, the release of monthly GRACE L2 product RL06 between 04/2002 to 09/2016 and the release RL05 between 11/2016 to 01/2017 in terms of spherical harmonic coefficients up to degree and order 96 were downloaded from the Center for Space Research (CSR, http://www2.csr.utexas.edu/grace/). GRACE-FO data were also downloaded from the CSR webpage. The combined monthly Swarm L2 gravity model was downloaded from http://www.asu.cas.cz/~bezdek/vyzkum/geopotencial/ in terms of spherical harmonic coefficients up to degree and order 40 between 12/2013 to 12/2018. The coefficients of degree one of GRACE(-FO) are augmented by those derived from Swenson et al. (2008), whereas the degree two coefficients are replaced by those derived from SLR data following Cheng et al. (2013). The degree one and two coefficients of the Swarm fields were also replaced to be consistent with the treatment of GRACE(-FO) processing. GIA corrections were applied after implementing the reconstruction. For details on the data processing and ICA approach see Forootan et al. (2020).

In this study, monthly GRACE(-FO) TWSA values are estimated on a regular global 0.5 deg grid. The grid values are spatially averaged over 148 river basins (TWSA validation basins). The TWSA validation basins were derived by combining a few of the 1509 streamflow calibration basins such that the area of each TWSA validation basin is larger than 200000 km$^2$. A two-step approach was applied to filter the observations and to compute and reduce leakage errors of the basin-averaged time series following the approach of Khaki et al. (2018). In the first step, a 2d-destriping filter was designed for the spectral domain that acknowledges the north-south striping pattern of the GRACE(-FO) error structure and aims to retain the high-frequency spatial changes while removing the noise. In the second step, an efficient averaging kernel was designed to spatially average the observations for the 148 selected river basins and simultaneously estimate the leakage-in and leakage-out of the signal. These estimates are used to correct the smoothed signal of step 1. The magnitude of the leakage error is used to represent the TWSA uncertainties because this error is dominant in the TWSA processing steps. We consider the time span between 01/2003 - 12/2019, limited by the common period of GRACE(-FO) data and model output from the different WaterGAP versions.

Note that we refer to the term "terrestrial water storage" specifically in context with GRACE(-FO). In contrast, the term "total water storage" remains in those cases where the context is with WaterGAP (e.g. the water balance assessments).

### 7.3 Evaluation metrics

The Nash-Sutcliffe efficiency metric $NSE$(-) (Nash and Sutcliffe, 1970), the Kling-Gupta efficiency metric $KGE$(-) with its components correlation $KGE_r$(-), bias $KGE_b$(-) and the deviation of variability $KGE_g$(-) (Kling et al., 2012; Gupta et al., 2009) as well as TWSA-related metrics are applied here and were described in Müller Schmied et al. (2021, their Sect. 6.3). To improve the readability of this paper, the definitions of the evaluation metrics are repeated in Appendix B.

### 7.4 Evaluation results

#### 7.4.1 Water abstractions

The evaluation of simulated potential abstractions against reported abstraction values in the AQUASTAT database (FAO, 2022) shows a reasonable model quality (Fig. 7). WaterGAP total withdrawal water uses and also total groundwater and surface withdrawals water use show a very good fit to the AQUASTAT data, which were not used as model input. Slightly less but still reasonable performance is shown for the sectors irrigation, industrial (manufacturing), domestic and thermoelectric. WaterGAP tends to overestimate withdrawal water uses in the industrial sector (Fig. 7e) and underestimate them in the domestic sector (Fig. 7f). The update of the thermoelectric and manufacturing sectors in WaterGAP v2.2e slightly decreases the fit to AQUASTAT data (comp. Figs. 7 and S8). In particular, the tendency of overestimation of withdrawal water uses in the thermoelectric sector in v2.2d is shifted also towards a partial underestimation in v2.2e. In addition, values for WaterGAP v2.2e are lower as compared to v2.2d. The distribution of the industrial sector in v2.2e tends to spread more as compared to v2.2d.

The performance of the livestock sector with an $NSE$ of 0.4 is relatively low and both overestimations and underestimations are visible (Fig. 7h). However, the total volumes are mostly below 1 $km^3yr^{-1}$ and the amount of data points from AQUASTAT is lowest among the other variables. The difference between the irrigation sector, and the corresponding total, groundwater and surface water withdrawal water uses due to the different climate forcings is rather low in comparison to AQUASTAT, as are the differences to WaterGAP v2.2d (Figs. S6-S9). A slightly lower fit of WaterGAP forced by ERA5 to AQUASTAT irrigation abstractions is observed (comp. Figs. 7 and S9).

#### 7.4.2 Streamflow

The evaluation of streamflow indicates the overall best results with WaterGAP v2.2e driven by gwsp3-w5e5 (Fig. 8 and Table 9). There are only very small differences between the model versions v2.2d and v2.2e under the same climate forcing. The gswp3-era5 climate forcing leads to a slightly lower performance with regard to mean bias ($KGE_b$) and variability ($KGE_g$). The simulations as driven by climate forcings that use 20crv3 prior to 1979 have much lower performance metrics than those that use gwsp3 (Figs. 8, D1). This is also visible in the cumulative distribution functions of $KGE$, $NSE$ and the KGE components (Figs. 9, D1, D2, D3, D4).

With WaterGAP v2.2e as driven by gswp3-w5e5, large areas of North America and Africa result in $NSE$ values below 0.5, which is a similar pattern as in Müller Schmied et al. (2021, their Fig. 7) (Fig. 10). Basins in the lowest $KGE$ class are the same as the basins with $NSE$ performance lower than 0.5 (Fig. 11a). As intended by the calibration routine, the $KGE_b$ is mostly around the value 1 (Fig. 11b). Deviations are due to a longer time series for evaluation for several stations and the model start in 1901 for evaluation instead of the calibration period (where time spans differ). There are many regions with close-to-optimal $KGE$ components $KGE_b$ and $KGE_r$ (Fig. 11c) but $KGE_g$ that deviates strongly from 1, indicating that streamflow variability is not simulated well (Fig. 11d). In most snow-dominated river basins, WaterGAP underestimates the variability. Correlations are poor in some dry and some snow-dominated basins. Performance in generally lower in highly anthropologically altered basins such as the outlet of the Nile basin where WaterGAP cannot simulate the seasonality and

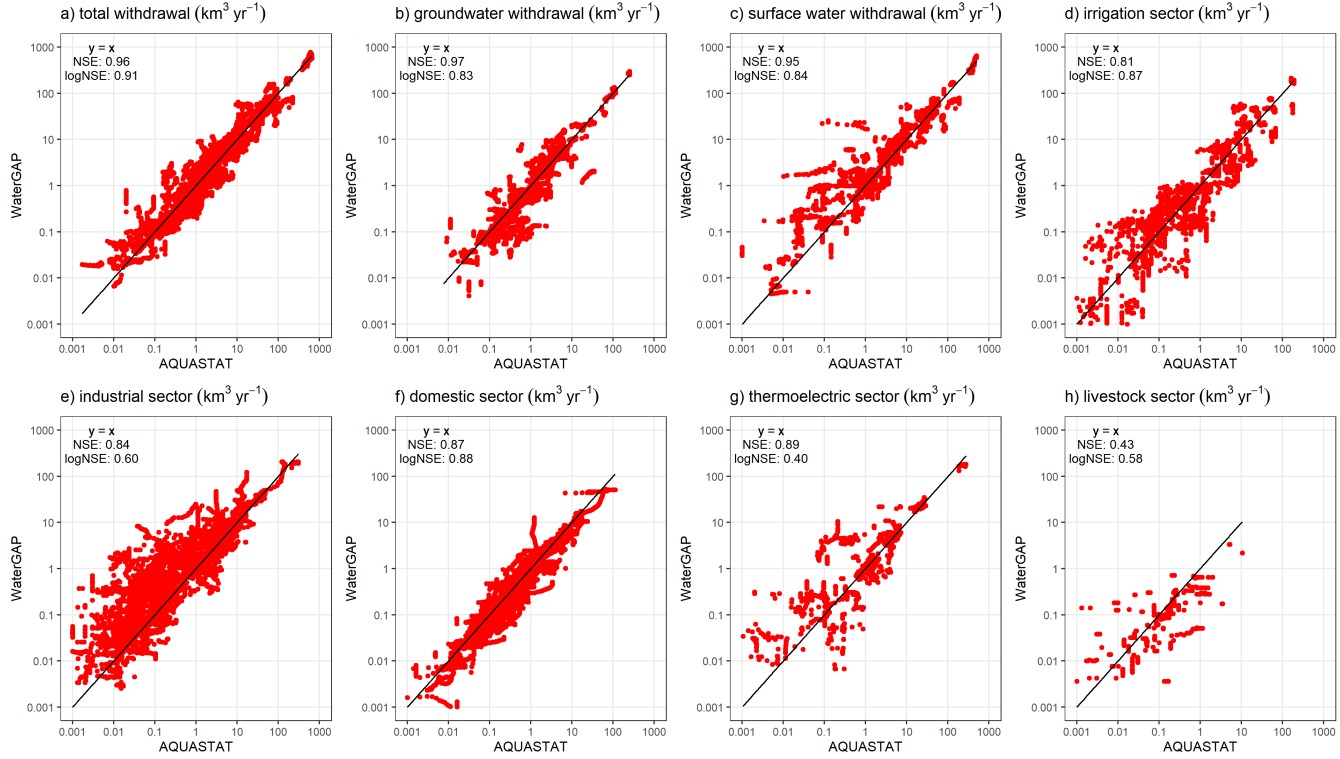

**Figure 7.** Comparison of potential withdrawal water uses from WaterGAP v2.2e and gswp3-w5e5 with AQUASTAT (FAO, 2022). Each data point represents one yearly value per country for the time span 1964-2019, if present in the database.

interannual variability of the upstream dam releases and water abstractions well, resulting in a low $KGE_r$ and $KGE_g$ (Fig. 11c, d).

Performances according to the Köppen-Geiger climate zones are shown in Tables 9, D3, D4, D2, D1. Please note that the assignment of a basin to the climate zone is based on the climate forcing used and can thus differ slightly among the model variants. When assessing the $KGE$ and $NSE$ performance indicator for Köppen-Geiger climate zones, a similar pattern is visible despite that the distribution in the classes is differing due to the obviously different meaning of the performance values (Table D1). Highest $KGE_r$ values are generally reached for A and C climates and especially here, the difference between the gswp3 and 20crv3 climate forcing combinations is visible (Table D2). For $KGE_b$, a tendency to simulate higher mean streamflow as compared to the observation is visible for A and C climates, whereas for the other climate zones, the number of basins is distributed rather equally around the 10% deviation that is introduced by the calibration routine (Table D3). The variability indicator $KGE_g$ differs largely from the optimum value, especially for A, B and D climate zones. For A (D) climates, all models underestimate variability around half (2/3) of the basins. The model variants as driven by ERA5 climate combinations have a tendency to underestimate variability, especially in C climates (Table D4).

The assessments above have been done using all monthly observation data available for the stations, including those monthly values that have not been used in model calibration. This dataset is referred to by "all data". The monthly data that was used (in yearly aggregation) for calibration is referred to by "calibration data". Finally, the difference of "all data" and "calibration data", i.e. the months that are not used for calibration is referred to as "validation data". A slight performance decrease occurs when evaluating the fit to simulated streamflow for a validation dataset mainly due to a reduced $KGE_b$ (see the corresponding Figures S11- S49 in the supplementary material).

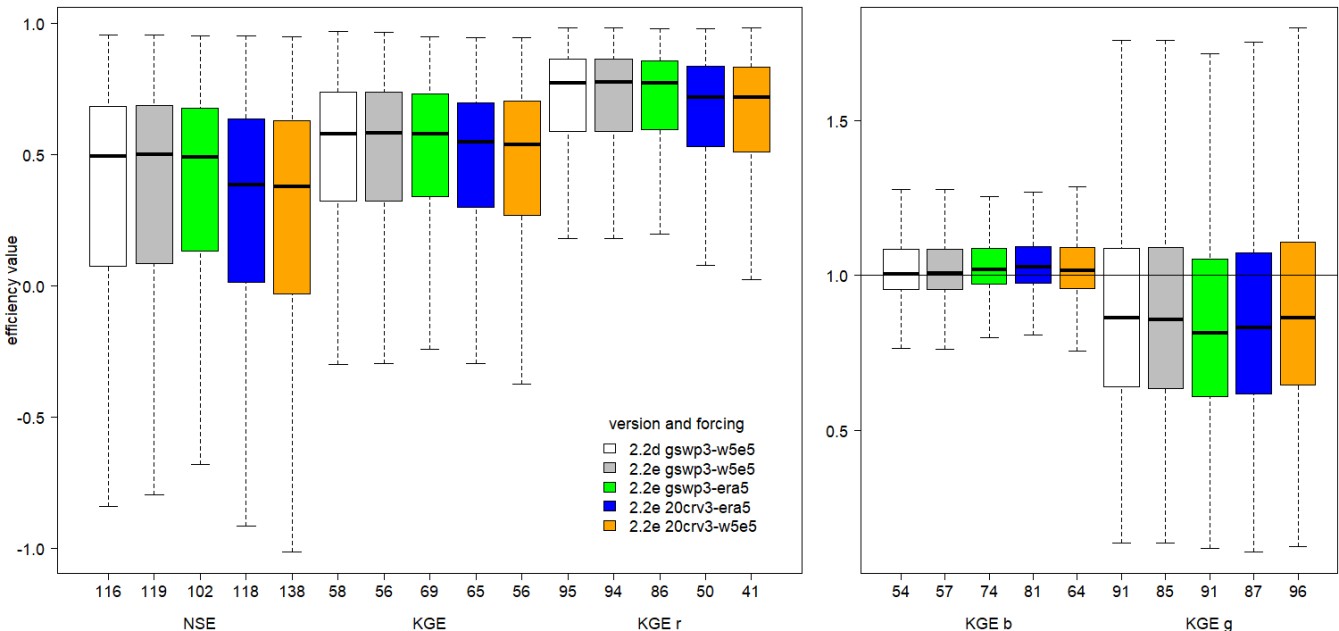

**Figure 8.** Efficiency metrics for monthly streamflow of the WaterGAP variants at the 1509 observation stations (all data) with $NSE$, $KGE$ and its components. Outliers (outside 1.5× inter-quartile range) are excluded but the number of stations that are defined as outliers are indicated at x-axis.

### 7.4.3 TWSA

The comparison of basin-average TWSA of WaterGAP v2.2e forced by gswp3-w5e5 and the reconstructed gap-free time series of GRACE(-FO) for 148 basins is shown in Fig. 12. The annual amplitude is underestimated in most of the African basins and in some Asian basins but is overestimated in major parts of North America. The correlation between WaterGAP v2.2e and GRACE(-FO) is overall reasonable with the majority of basins experiencing correlations between 0.5-1. However, basins where the amplitude is considerably under- or overestimated show low correlations. The comparison of TWSA trends shows that WaterGAP v2.2e generally computes considerably smaller trends in comparison to GRACE(-FO). This characteristic was also observed in the previous model evaluation (Müller Schmied et al., 2021).

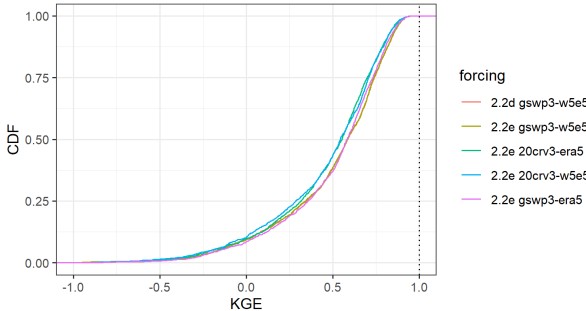

**Figure 9.** Cumulative distribution of the $KGE$ efficiency metric for all monthly streamflow values at the 1509 gauging stations for all model variants.

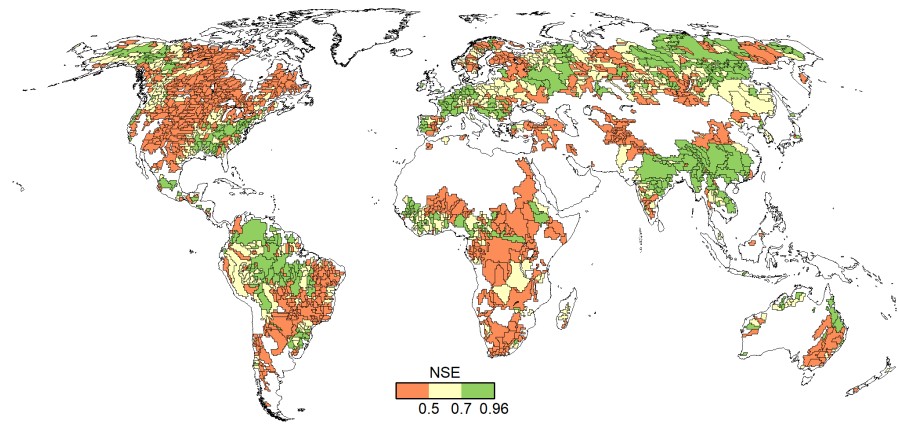

**Figure 10.** $NSE$ efficiency metric for all monthly data of the 1509 river basins in WaterGAP v2.2e as forced by gswp3-w5e5.

The comparison between WaterGAP v2.2d and v2.2e shows that only a few basins differ; mainly stronger trends in (north-)east Asia can be observed for version v2.2e. The WaterGAP v2.2e versions forced by 20crv3-era5 and gswp3-era5, respectively, show only marginal differences. This is expected since both versions are forced by ERA5 during the evaluation period for TWSA (01/2003-12/2019). When forcing the model with ERA5, stronger trends are observed in North America than with W5E5. The correlations differ in (north)-east Asia, and match better in South America. The annual amplitude fits better in North America, but the annual amplitude in South America is better represented using the W5E5 forcing.

## 7.5 Performance changes due to the updated calibration data basis

The calibration data basis with observed mean annual streamflow values of WaterGAP v2.2e has 190 stations more than WaterGAP v2.2d. In particular, 77 river basins are newly included in the calibration routine (ID 1). In 6 cases, a new gauging station has been added downstream (ID 2) and, in 126 cases, upstream (ID 3) of an already existing station. For 21 basins, a

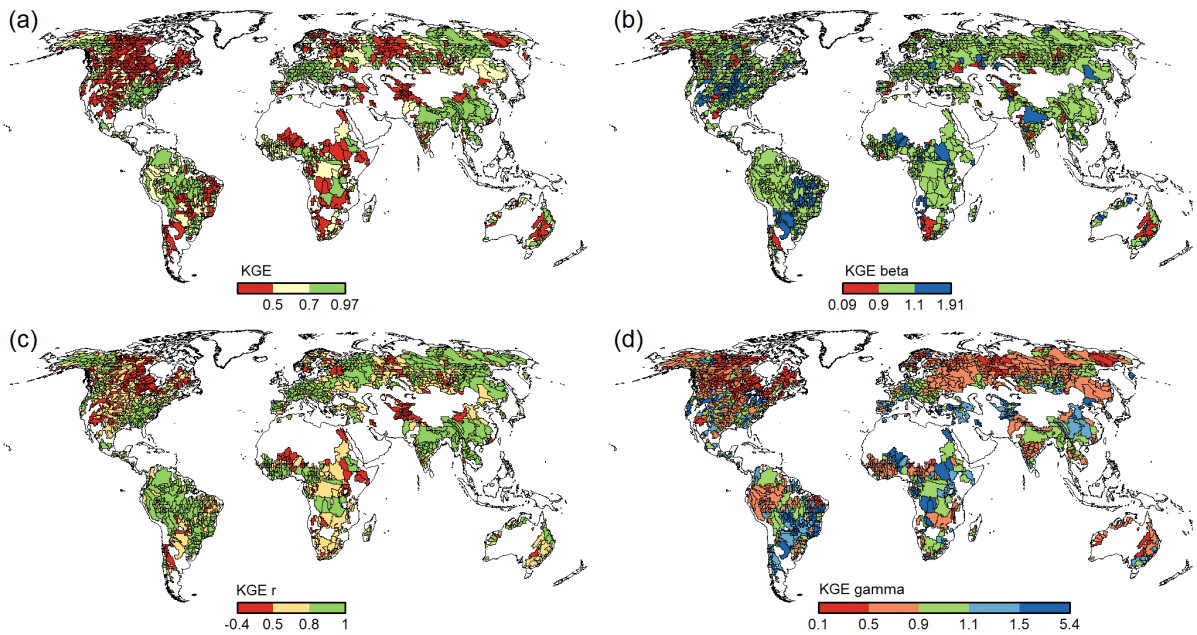

**Figure 11.** $KGE$ efficiency metric and its components for all monthly streamflow values at the 1509 gauging stations for WaterGAP v2.2e as forced by gswp3-w5e5.

station was moved as compared to the previous calibration data basis (ID 4). These sums up to 230 gauging stations that differ between the calibration data basis of v2.2d and v2.2e.

To determine the impact of the updated streamflow data basis, the performance of simulated streamflow obtained by calibrating WaterGAP v2.2d against the two different streamflow datasets (1319 vs. 1509) was compared for the 230 stations. Due to the similar performance between the two model versions, we do not expect that analysis results with v2.2e would be similar. The gswp3-w5e5 climate forcing was applied in both variants.

For all 230 stations, the calibration with the updated observational data basis, which is used to calibrate the standard version of WaterGAP v2.2e, led to substantially improved performance indicators, in particular $NSE$, $KGE$ and $KGE_b$, whereas $KGE_r$ and $KGE_g$ do not differ notably (Fig. 13). This improvement is a result of the calibration objective to adjusts the bias in mean simulated streamflow to a range of 10 % around the observed value.

Strong performance improvements are observed for the 77 grid cells with newly added calibration data that are outside (and also not downstream) of previously calibrated basins (ID 1), considering both the median as well as the spread (indicated by the range of the 25th and 75th percentile) (Tab. 10). Those grid cells that are already calibrated by a more downstream station in the case of the old calibration data basis (ID 3) show less performance gain. In particular, $KGE_b$ for the ID 3 stations is already close to the optimum value due to calibration to a downstream observation. Here, the bias adjustment of the downstream station is effective for upstream grid cells. In contrast, the improvement is large if stations are included further downstream of an already existing station (ID 2) but the small number of stations implies a careful interpretation (Tab. 10).

**Table 9.** Number of calibration basins in each Köppen-Geiger region for which $KGE$ of monthly streamflow time series is within three performance classes, for five WaterGAP variants. Note that the assignment of a basin to a climate region can differ among the climate forcings.

| Model variant | $KGE$ | A | B | C | D | E | sum |
|---|---|---|---|---|---|---|---|
| | >0.7 | 127 | 17 | 163 | 167 | 15 | 489 |
| 22d gswp3-w5e5 | 0.5-0.7 | 124 | 37 | 77 | 173 | 12 | 423 |
| | <0.5 | 109 | 72 | 68 | 329 | 19 | 597 |
| | >0.7 | 127 | 17 | 163 | 168 | 15 | 490 |
| 22e gswp3-w5e5 | 0.5-0.7 | 125 | 38 | 77 | 175 | 13 | 428 |
| | <0.5 | 108 | 71 | 68 | 326 | 18 | 591 |
| | >0.7 | 78 | 6 | 105 | 170 | 11 | 370 |
| 22e 20crv3-era5 | 0.5-0.7 | 137 | 35 | 102 | 186 | 9 | 469 |
| | <0.5 | 133 | 76 | 114 | 339 | 8 | 670 |
| | >0.7 | 96 | 8 | 111 | 159 | 15 | 389 |
| 22e 20crv3-w5e5 | 0.5-0.7 | 129 | 37 | 93 | 190 | 5 | 454 |
| | <0.5 | 132 | 83 | 106 | 326 | 19 | 666 |
| | >0.7 | 96 | 7 | 152 | 173 | 13 | 441 |
| 22e gswp3-era5 | 0.5-0.7 | 142 | 38 | 102 | 207 | 8 | 497 |
| | <0.5 | 112 | 70 | 70 | 310 | 9 | 571 |

**Table 10.** Model performance for the two calibration variants (1509 vs. 1319 stations) and the ID[1] with reason for change between the two variants and the corresponding number of affected stations in brackets. The performance indicator is provided as median with its 25th and 75th percentile in brackets.

| ID[1] | variant | $NSE$ | $KGE$ | $KGE_r$ | $KGE_b$ | $KGE_g$ |
|---|---|---|---|---|---|---|
| 1 (77) | 1509 | 0.37 (-0.07 \| 0.68) | 0.58 (0.19 \| 0.73) | 0.75 (0.55 \| 0.87) | 1.00 (0.93 \| 1.09) | 1.01 (0.78 \| 1.19) |
| | 1319 | -0.31 (-4.89 \| 0.40) | 0.00 (-0.77 \| 0.49) | 0.78 (0.57 \| 0.87) | 1.39 (0.89 \| 2.61) | 1.00 (0.75 \| 1.32) |
| 2 (6) | 1509 | 0.55 (0.19 \| 0.83) | 0.54 (0.43 \| 0.81) | 0.75 (0.51 \| 0.92) | 1.01 (0.94 \| 1.05) | 0.93 (0.67 \| 1.07) |
| | 1319 | -0.27 (-1.05 \| 0.61) | 0.08 (-0.44 \| 0.69) | 0.76 (0.50 \| 0.91) | 1.69 (1.08 \| 2.39) | 0.91 (0.81 \| 1.03) |
| 3 (126) | 1509 | 0.15 (-0.26 \| 0.61) | 0.44 (0.03 \| 0.69) | 0.73 (0.34 \| 0.85) | 1.02 (0.97 \| 1.09) | 0.85 (0.59 \| 1.31) |
| | 1319 | -0.03 (-0.97 \| 0.44) | 0.19 (-0.14 \| 0.58) | 0.71 (0.35 \| 0.85) | 1.04 (0.82 \| 1.39) | 0.86 (0.62 \| 1.29) |
| 4 (21) | 1509 | 0.55 (0.15 \| 0.69) | 0.62 (0.49 \| 0.78) | 0.77 (0.62 \| 0.88) | 1.00 (0.94 \| 1.09) | 0.89 (0.81 \| 1.15) |
| | 1319 | 0.18 (-0.34 \| 0.60) | 0.45 (0.31 \| 0.68) | 0.80 (0.57 \| 0.87) | 1.18 (0.98 \| 1.45) | 0.93 (0.84 \| 1.23) |

[1] 1: new river basin, 2: added station downstream of already existing station, 3: added station upstream of already existing station, 4: station that was moved.

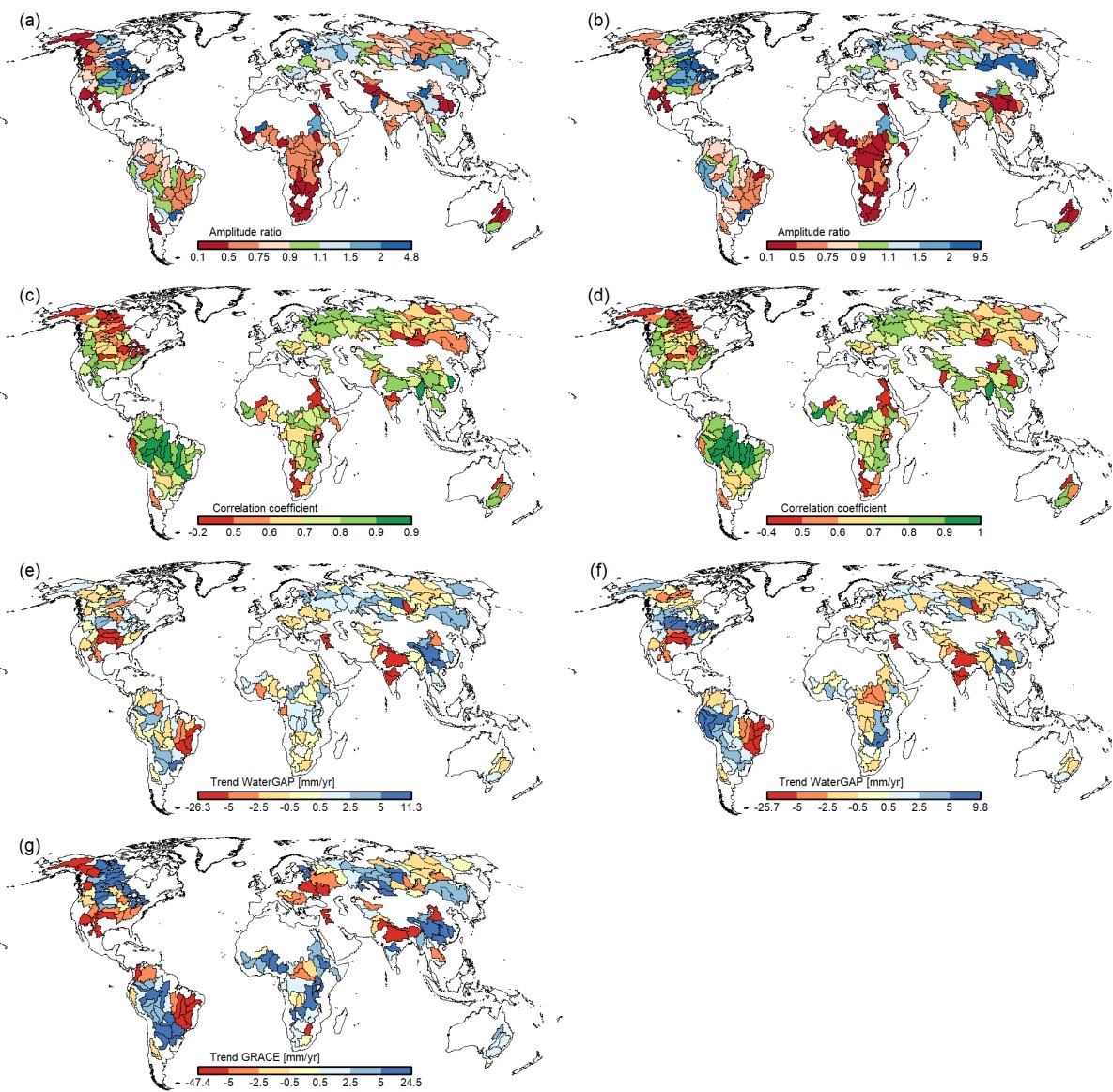

**Figure 12.** Comparison of basin-average monthly time TWSA time series of WaterGAP v2.2e as forced by gswp3-w5e5 (a, c, e) and gswp3-era5 (b, d, f) for 148 basins larger than 200,000 $km^2$, with (a, b) ratio of amplitude (reddish colors indicate amplitude underestimation by WaterGAP), (c, d) correlation coefficient, (e, f) trend of WaterGAP v2.2e and (g) trend of GRACE. All values are based on the time series January 2003 to December 2019.

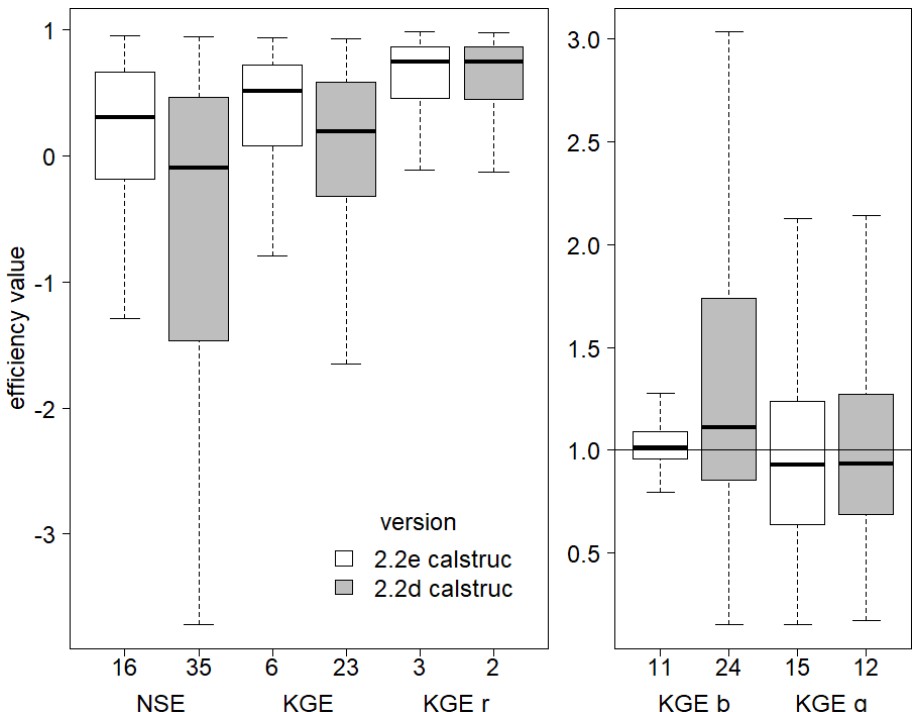

**Figure 13.** Efficiency metrics for monthly streamflow of the 230 gauging stations that differ between the streamflow data basis used for calibrating WaterGAP v2.2d and the new data basis used for v2.2e., with $NSE$, $KGE$ and its components. All monthly observations available have been used to compute the metrics. Outliers (outside 1.5× inter-quartile range) are excluded but the number of stations that are defined as outliers is indicated at x-axis.

## 7.6 Performance comparison between different model variants

### 7.6.1 WaterGAP v2.2e vs. WaterGAP v2.2d

The performance of simulated water abstractions is nearly identical except for the thermoelectric sector, where WaterGAP v2.2e, with the updated water use, results in a slightly worse fit to AQUASTAT data (logarithmic $NSE$ is 0.40 for v2.2e and 0.52 for v2.2d) (Figs. 7 and S8). With regards to the streamflow performance, WaterGAP 2.2e performs nearly identically as WaterGAP v2.2d with the same climate forcing and calibration data. This is also visible in the spatial pattern for streamflow where differences are rare. The performance ratio of indicators (for calculation see the Appendix C) often shows basins with
slightly different sign next to each other (Fig. 14) but without a clear spatial pattern of general performance gain or loss. When aggregated to climatic characteristics, such as Köppen-Geiger regions, it can be seen that WaterGAP v2.2e has slightly more basins in a better $KGE$ class for cold D and E climate as compared to WaterGAP v2.2d with the same climate forcing (Table 9).

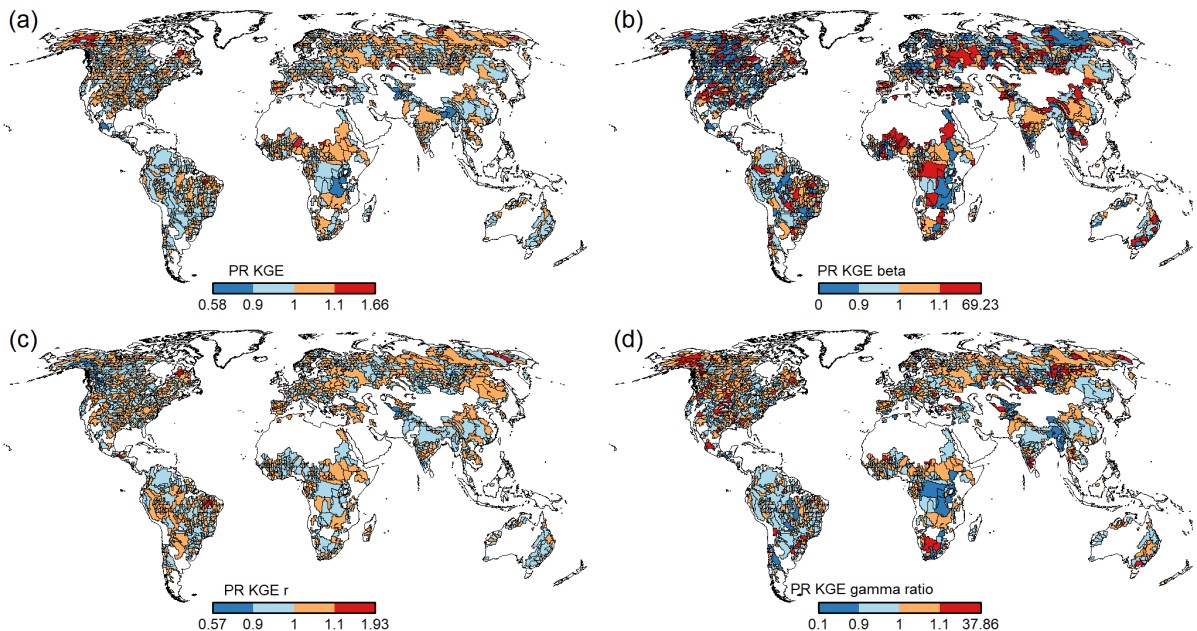

**Figure 14.** Resulting performance ratio of indicators of streamflow for the model version v2.2d and v2.2e as driven by gswp3-w5e5 for overall KGE (a), KGE beta (b), KGE correlation r (c) and KGE variability gamma (d). Bluish colours indicate that v2.2e is closer to the optimal parameter indicator value than v2.2d, see also the description in Appendix C. Note that the calibration procedure forces KGE beta values to be close to the optimum value, hence the drastic colours here are a result of only small differences to the optimum value.

For TWSA WaterGAP v2.2e performs better than v2.2d, specifically as the trends (in both directions) of TWSA are stronger for v2.2e and fit better to the observations, but also correlation coefficients as well as the amplitude ratios are improved for v2.2e. The performance ratio of indicators for TWSA shows for most basins a consistent direction of change for trend and correlation (with more bluish colours, indicating more regions with performance gain with v2.2e) while the amplitude sometimes show the opposite signal, especial for those regions with an improved trend ratio (Fig. 15). The seasonality of streamflow and TWSA is rather similar within the 12 selected river basins (Fig. S54).

### 7.6.2 GSWP3-W5E5 vs. GSWP3-ERA5

The impact of the selected climate forcing starting in 1979 is substantial, except for the water use (where the performance of gswp3-era5 regarding irrigation water abstractions is slightly lower).

The median streamflow performance with gswp3-w5e5 is slightly higher than with gswp3-era5 (value in brackets) with 0.499 (0.490) for $NSE$, 0.582 (0.578) for $KGE$, 0.775 (0.774) for $KGE_r$, 1.007 (1.018) for $KGE_b$ and 0.858 (0.813) for $KGE_g$. In particular, the Köppen climate zone A (equatorial climate) shows higher performance with gswp3-w5e5 (Table 9). Model simulations driven by ERA5-combinations have higher $NSE$ values in northwestern North America but lower values in China (comp. Figs. 10, and S33). But also, ERA5 combinations tend to have a lower $KGE_r$ in some parts of North America

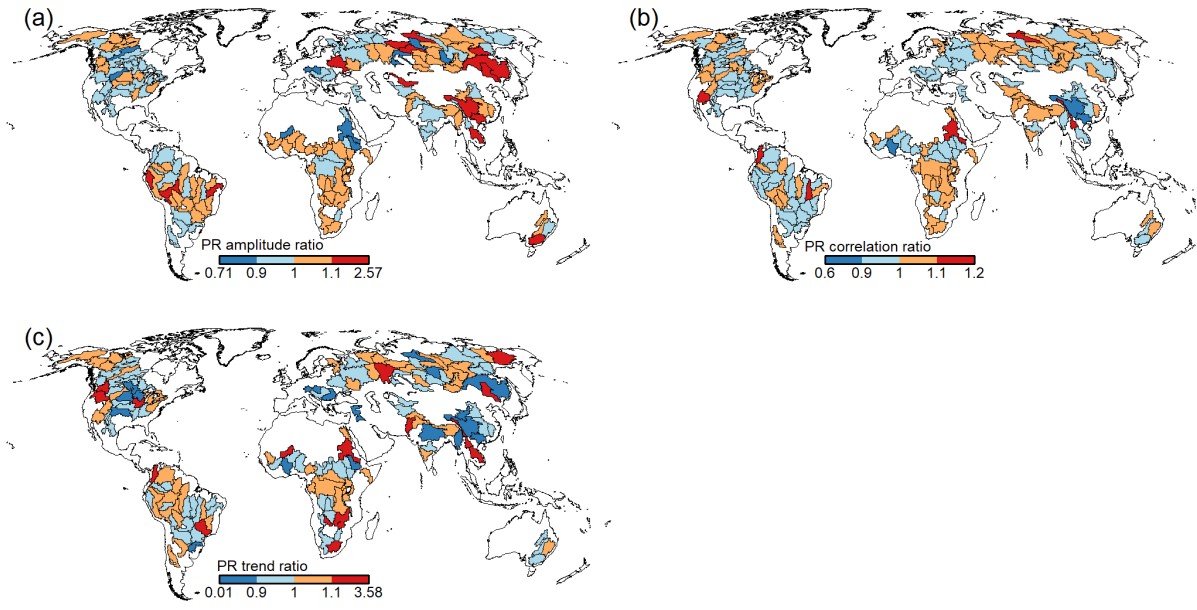

**Figure 15.** Resulting performance ratio of indicators of TWSA for the model version v2.2d and v2.2e as driven by gswp3-w5e5 for the amplitude ratio (a), correlation ratio (b) and trend ratio (c). Bluish colours indicate that v2.2e is closer to the optimal parameter indicator value than v2.2d, see also the description in Appendix C.

and large parts of South America and a generally higher variability as compared to the W5E5 combinations (comp. Figs. 11 and S41).

The TWSA trend in gswp3-era5 is closer to the observations in North and South America and the amplitude ratio is also improved for North America. For parts of Europe and Asia, the correlation but also the trend as driven by gswp3-w5e5 are closer to GRACE, showing an overall diverse impact of climate forcing to TWSA (Fig. 12). This is also visible in the seasonality, where large differences occur both for streamflow and for TWSA (Fig. S55). For example, the TWSA as driven by gswp3-era5 matches perfectly to observations for the Amazon but for streamflow, gswp3-w5e5 fits better.

### 7.6.3   GSWP3-W5E5 vs. 20CRv3-W5E5

Performance metrics for water abstraction are identical for both variants (Figs. 7 and S10). The median streamflow performance with gswp3-w5e5 is generally higher than with 20crv3-w5e5 (value in brackets) with 0.499 (0.378) for $NSE$, 0.582 (0.539) for $KGE$, 0.775 (0.718) for $KGE_r$, 1.007 (1.015) for $KGE_b$ except for $KGE_g$ with 0.857 (0.864). The higher performance of gswp3-w5e5 is obvious for all Köppen-climate regions, with smaller differences for D and E climates (Table 9. Differences
in seasonality are relatively small as the time series for TWSA and streamflow starts several years after 1979 and thus use both W5E5. The visible differences are related to the specific calibration parameters that depend also on the years before 1979.

## 8    Benefits and limitations of the calibration approach

The calibration of WaterGAP is a simple but effective approach to adjust biases in simulated streamflow, runoff and renewable water resources. As shown for the 230 grid cells with new streamflow observations used for calibrating WaterGAP v2.2e, calibration leads to an overall reduction of water resources to be closer to the observations (Table 10). Previous assessments of WaterGAP determined that the decision to calibrate or not has the largest effect on water resources, both on global scale-fluxes but also at the spatial runoff pattern (Müller Schmied et al., 2014). The improved representation of long-term average water resources is required for evaluating water stress. In addition, this bias adjustment, which also balances out uncertainties in precipitation, is beneficial for improving the simulation of, e.g., the dynamics of downstream wetlands or reservoirs.

However, the simple approach to modify only one parameter ($\gamma$) and up to 2 additional correction factors by calibration against mean annual streamflow has limitations. Reaching the calibration objective by modifying $\gamma$ alone is possible only in 519 (524) basins of WaterGAP v2.2e (v2.2d), which indicates that the uncertainties in input data model structure and the many other model parameters might not be covered well by adjusting only this parameter. In most of the other basins, runoff is still overestimated with the optimum $\gamma$, and the correction factors need to lower the runoff. Another model parameter, the maximum soil water storage $S_{max}$, has been found to strongly affect runoff generation and the seasonality and trends of terrestrial water storage anomalies (Tangdamrongsub et al., 2018; Scanlon et al., 2019), with higher values decreasing runoff and increasing seasonality and trends. Multi-variable calibration of WaterGAP in individual basins (Hosseini-Moghari et al. (2020); Döll et al. (2024) and comparison of model output to spaceborne terrestrial water storage anomalies indicates that the cell-specific $S_{max}$ values used in WaterGAP might be too low. Thus, increased $S_{max}$ values are expected to help achieve the calibration objective by adjusting $\gamma$ alone.

More complex multi-variable calibration approaches, which use not only observed streamflow but also observations of other model output variables such as TWSA or snow cover, allow us to go beyond bias adjustment and adjust more model parameters. While such ensemble-based calibration approaches have been successfully applied to WaterGAP for individual basins such as Mississippi sub-basins (Döll et al., 2024), they are not yet applicable as a standard approach for global-scale calibration. Such ensemble-based calibration approaches are computationally expensive and also suffer from methodological problems related, for example, to the large footprint of spaceborne terrestrial water storage anomalies (>100,000 $km^2$) or trade-offs between the optimal simulation of the different observed variables (Döll et al., 2024).

## 9    Standard model output

Similar to Müller Schmied et al. (2021), we provide standard output data for WaterGAP v2.2e driven by the four climate forcings listed in Table 1 and for comparison also WaterGAP v2.2d driven by gwps-w5e5. In addition to the standard "ant" runs that include direct human impacts (water use and man-made reservoirs, labeled with $histsoc$), we provide, for all five variants, the model output of "nat" model runs where it is assumed that there is no human water use and no man-made reservoirs (labeled with $nosoc$). The data are stored using the network Common Data Form (netCDF) format developed by UCAR/Unidata (Rew et al., 1989) and are available at Goethe University Data Repository (GUDe) (Müller Schmied et al.,

2023a, b, c, d, e, f, g, h, 2024a, b). For two forcings and the "ant" runs, daily temporal resolution for the storage compartments are provided (Müller Schmied et al., 2024c, d). The netCDF files contain metadata with detailed information regarding characteristics of the data, e.g., whether a storage type contains anomaly or absolute values and a legend where applicable.

The available water storages, flows and water use variables are listed in Tables E1, E2 and E3 respectively. Table E4 includes additional data, such as the cell-specific continental area as used in WaterGAP v2.2e to convert between equivalent water
heights (e.w.h.) and volumetric units (assuming a water density of $1 \, \mathrm{g\,cm^{-3}}$). A spatial view for a range of model output is available in a Web-App (Attard, 2024).

## 10  Caveats of WaterGAP v2.2e

This section is a compilation of known issues with the model output and should give guidance to data users.

- Due to the architecture of WaterGAP, where the output of individual water use models are combined to net abstractions
from groundwater and net abstractions from surface water in the linking model GWSWUSE (Müller Schmied et al., 2021, their Sect. 3.3), it is not possible to compute sectoral actual consumptive water use values (and the corresponding withdrawal water uses) but only total actual consumptive water use (and corresponding withdrawal water use).

- In WaterGAP, the actual total consumptive water use (variable atotuse) is added to the actual evapotranspiration (evap). In cases where surface water abstractions are satisfied from the neighbouring cell due to shortages in the original water-
demanding cell, the return flows to groundwater are assigned to the original water-demanding cell. This can lead to 1) a negative value for atotuse and 2) in those cases where the (positive) evap value are low, to a in total negative evap value.

- In dry areas around large rivers, water is often abstracted from neighbouring cells with big rivers (e.g., the Nile) to satisfy water demand in the original demand cell. The return flows are increasing the groundwater in the demanding cell, which results in a relative increase of groundwater storage and thus an increase of groundwater outflow which is then visible in
the total runoff qtot and could be in sum more than the precipitation (precip) in the grid cell. Furthermore, the calibration factor CFA can lead to more runoff than precipitation.

- When comparing globally aggregated streamflow from previous versions with WaterGAP v2.2e it has to be considered that due to the new handling of inland sinks in WaterGAP v2.2e (Sect. 2.5) the endorheic basins contribute to actual evaporation and the sink cells have zero streamflow. When quantifying the renewable water resources on the global
scale, inflow to to all inland sinks has to be added to the water resources of the other cells (or the streamflow into oceans).

## 11  WaterGAP v2.2e in ISIMIP3

WaterGAP contributes to the Inter-Sectoral Impact Model Intercomparison Project (ISIMIP) in its current project phase 3 and follows the simulation protocol of ISIMIP (2023c). The model dashboard is available at ISIMIP (2023a) and an overview of

750 the simulated scenarios at ISIMIP (2023b). Model output can be accessed at ISIMIP (2023b). Mainly due to the architecture of WaterGAP, the following deviations from the simulation protocol exist:

- The drainage direction map used in WaterGAP does not completely follow the ISIMIP land-sea mask definition, which was modified slightly and unintendedly. In particular, the lat/lon 178.75, -49.25 (an island south-east of New Zealand) is defined as land but the drainage direction map used in WaterGAP locates this island in a neighbouring cell. Thus, this

island is not present and any model output for the grid cell with lat/lon 178.75, -49.75 is set to missing value in all files prepared for ISIMIP.

- The WaterGAP drainage direction map differs in four grid cells at Lake Ladoga in the Neva river basin in Russia from the ISIMIP definition (lat/lon coordinates 61.25, 31.25; 60.75, 31.25; 60.75, 31.75; and 60.75, 32.25). Those grid cells are not included in WaterGAP and drainage direction flows around this lake, resulting in a total number of 67420 grid

cells considered in WaterGAP v2.2e.

- WaterGAP does not use the land use data as provided by ISIMIP but a static, satellite-based map of land cover classes (Müller Schmied et al., 2021, their Appendix C). WaterGAP considers temporally varying irrigation areas (Müller Schmied et al., 2021, their Sect. 3.1) but not from ISIMIP.

- During the update of the reservoir data (Sect. 2.2) we found better-suited grid cell locations for several dams as compared

to the input data provided by ISIMIP. The data used within WaterGAP v2.2e is available via Müller Schmied and Trautmann (2023).

- According to the modeling protocol, the variable *qtot* consists of the sum of surface *qs* and sub-surface *qsb* runoff and is defined as total runoff. However and specifically for WaterGAP, this implies that for *qtot* (but not for the net cell runoff *ncrun* provided in the standard model output), the horizontal water balance (i.e., the water balance of the surface water

bodies) is not considered. For users who want to assess the differences, we provide *qtot* and *ncrun* as standard model output.

## 12 Conclusions and outlook

Since the development of the WaterGAP model started in 1996, numerous model versions have been created and applied in many studies. This paper describes the most recent model version v2.2e as well as the model output, with a focus on the

775 changes from the previous model version v2.2d described in Müller Schmied et al. (2021). With version v2.2e, the applicability of WaterGAP for answering scientific questions has been enhanced as compared to previous versions. The performance of v2.2e regarding water use, streamflow and TWSA do not differ much from v2.2d when using the same climate forcing and the same streamflow observations for model calibration (thus the only difference is to the model structure). The climate forcing gswp3-w5e5 leads to the highest performance for streamflow, whereas there are distinct regions where gswp3-era5 is superior

to gswp3-w5e5 in particular for TWSA trends.

While version v2.2e has been finalized, the scientific and societal demand for future model development remains. For example, to improve the still poor simulation of the outflow and storage dynamics of artificial reservoirs, the reservoir algorithm should be modified and calibrated, benefiting from the recent availability of remote sensing based estimates of reservoir water storage dynamics. The achieved glacier integration into WaterGAP 3.2, which has led to an improved representation of TWSA (Cáceres et al., 2020), is unsustainable in the sense that it depends on updates from the glacier modeling community. Therefore, model adjustments and arrangement with the glacier modeling community are required to achieve a continuing integration of glacier model output into WaterGAP, which would particularly improve climate change impact assessments (Hanus et al., 2024). Then, a future model version of WaterGAP could include a glacier component in its standard variant.

The WaterGAP v2.2e software, written in C/C++, started to be developed nearly 30 years ago. Generations of researchers modified, tested and documented the code, resulting in a very complex software that is difficult to understand, maintain and enhance. Currently, the WaterGAP Global Hydrology Model and GWSWUSE are re-programmed in Python, with a modern software architecture; this research software will be available as an open source community software alongside documentation, user guide and examples (Nyenah, 2024).

*Code and data availability.* The code of WaterGAP v2.2e is open source under GNU Lesser General Public License version 3 license at Müller Schmied et al. (2023i). The model output data availability is described in Sect. 9. The streamflow data for the evaluation is available at Müller Schmied and Schiebener (2022), the GRACE(-FO) data are available at Forootan et al. (2020). For latest papers published based on WaterGAP 2, we refer to http://www.watergap.de, last access: 20 September 2023.

*Author contributions.* HMS and PD led the development of WaterGAP v2.2e. HMS led the software development, supported by TT, SA, DC, TAP, CH, PD. The paper was conceptualized by HMS and PD. HMS did the calibrations, simulations, data analysis, prepared the model output for the GUDe data repository, did the visualization and model validation, supported by MS regarding validation against GRACE TWSA. EK provided the updated non-irrigation water use data. The original draft was written by HMS, with specific parts drafted by TT, SA, DC, MF, HG, TAP, LS, MS and PD. All authors contributed to the final draft.

*Competing interests.* The authors declare that they have no conflict of interest.

*Acknowledgements.* We acknowledge the ISIMIP team for producing, and making available the ISIMIP input data. We thank Georg Seitfudem for support in finding and solving the bug in domestic water use data. We furthermore thank Lukas Grittner for polishing the reference list and for technical support during manuscript preparation. We thank Seyed-Mohammad Hosseini-Moghari for reviewing the draft. We are grateful to Guillaume Attard for creating the WaterGAP Web Explorer. We thank for valuable comments and suggestions from two anonymous referees which helped to streamline and improve the consistency of the paper.

## Appendix A: Technical changes

- Output of monthly groundwater recharge below surface water bodies is now possible.

- Data arrays are now stored and processed in std::vector objects.

- Several options to run WaterGAP were removed because they were not used anymore.

- Bug in the initialization of reservoir water demand in the respective commissioning years was fixed (routing routine).

- Bug in reintroduction of return flows into groundwater due to delayed satisfaction of $NA_S$ was fixed.

- Bug in reallocation of unsatisfied $NA_S$ at global lakes and reservoirs was fixed.

## Appendix B: Evaluation metrics

The following section is to large parts identical Müller Schmied et al. (2021, their Sect. 6.3) but repeated here for better readability of this paper.

## B1 Nash-Sutcliffe Efficiency

The Nash-Sutcliffe efficiency metric $NSE$ $(-)$ (Nash and Sutcliffe, 1970) is a traditional metric in hydrological modeling. It provides an integrated measure of model performance with respect to mean values and variability and is calculated as:

$$NSE = 1 - \frac{\sum_{i=1}^{n}(O_i - S_i)^2}{\sum_{i=1}^{n}(O_i - \overline{O})^2} \tag{B1}$$

where $O_i$ is observed value (e.g., monthly streamflow), $S_i$ is simulated value and $\overline{O}$ is mean observed value. The optimal value of $NSE$ is 1. Values below 0 indicate that the mean value of observations is better than the simulation (Nash and Sutcliffe, 1970). For assessing the performance of low values of water abstraction (Sect. 7.4.1), a logarithmic $NSE$ was calculated in addition by applying logarithmic transformation before calculation of the performance indicator.

## B2 Kling-Gupta Efficiency

The Kling-Gupta efficiency metric $KGE$ (Kling et al., 2012; Gupta et al., 2009) transparently combines the evaluation of bias, variability and timing and is calculated (in its 2012 version) as:

$$KGE = 1 - \sqrt{(KGE_r - 1)^2 + (KGE_b - 1)^2 + (KGE_g - 1)^2} \tag{B2}$$

where $KGE_r$ is the correlation coefficient between simulated and observed values $(-)$, an indicator for the timing, $KGE_b$ is the ratio of mean values (Eq. (B3)) $(-)$, an indicator of biases regarding mean values and $KGE_g$ is the ratio of variability (Eq. (B4)) $(-)$, an indicator for the variability of simulated ($S$) and observed ($O$) values.

$$KGE_b = \frac{\mu_S}{\mu_O} \tag{B3}$$

$$KGE_g = \frac{CV_S}{CV_O} = \frac{\sigma_S/\mu_S}{\sigma_O/\mu_O} \tag{B4}$$

where $\mu$ is mean value, $\sigma$ is standard deviation and $CV$ is coefficient of variation. The optimal value of $KGE$ is 1.

## B3 TWSA-related metrics

For the evaluation of TWSA performance, the following metrics were used: $R^2$ (coefficient of determination) as strength of linear relationship between simulated and observed variables, the amplitude ratio as indicator for variability and trend of both GRACE and WaterGAP data. Amplitude and trends were determined by a linear regression for estimating the most dominant temporal components of the GRACE time series. The time series of monthly TWSA was approximated by a constant $a$, a linear trend $b$, an annual and a semi-annual sinusoidal curve as follows

$$y(t) = a + b*t + c*sin(2*\pi*t) + d*cos(2*\pi*t) + e*sin(4*\pi*t) + f*cos(4*\pi*t) + r \tag{B5}$$

where $r$ denotes the residuals. The parameters $a$ to $f$ were estimated via least-squares adjustment. The annual amplitude can be computed by $A = sqrt(c^2 + d^2)$, and thus, the annual ratio was calculated by $A_{WGHM}/A_{GRACE}$.

## Appendix C: Performance ratio of indicators

In order to find out where the difference to the optimal value of a model performance indicator is reduced or increased between the two versions (v2.2e vs. v2.2d) of WaterGAP, the indicator performance ratio (Eq. C1) was used, defined as:

$$PR_{IND} = \frac{|1.0 - IND_{v2.2e}|}{|1.0 - IND_{v2.2d}|} \tag{C1}$$

where: $PR_{IND}$ is the performance ratio of the given indicator $IND$ [-], $IND$ is the indicator value (KGE and its components for streamflow; Amplitude ratio for TWSA and the ratio of Model divided by GRACE for TWSA trend) for the particular model version [-]. The smaller the resulting $PR_{IND}$ is, the better v2.2e is compared to v2.2d. For $PR_{IND}$ values < 1.0, v2.2e performs better than v2.2d and vice versa. The closer $PR_{IND}$ is to 0, the better performs v2.2e against v2.2d.

## Appendix D: Additional figures and tables

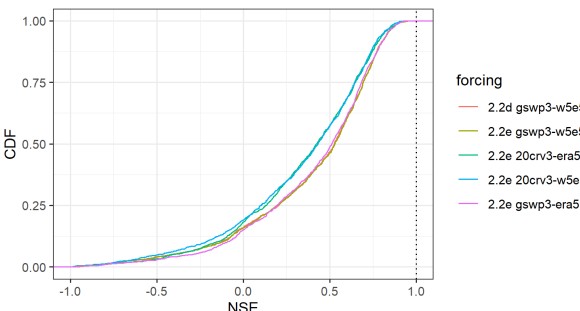

**Figure D1.** Cumulative distribution of the $NSE$ efficiency metric for all streamflow values at the 1509 gauging stations for all model variants.

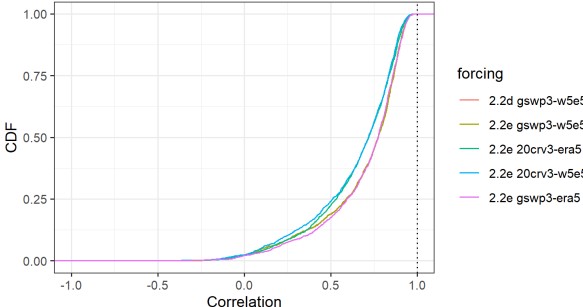

**Figure D2.** Cumulative distribution of the $KGE$ efficiency metric (correlation parameter) for all streamflow values at the 1509 gauging stations for all model variants.

**Table D1.** Model performance and the $NSE$ efficiency indicator and number of basins per Köppen-Geiger region in the particular performance class for the different WaterGAP variants.

| Model variant | $NSE$ | A | B | C | D | E | sum |
|---|---|---|---|---|---|---|---|
| | >0.7 | 87 | 13 | 112 | 109 | 14 | 335 |
| 22d gswp3-w5e5 | 0.5-0.7 | 114 | 25 | 83 | 183 | 6 | 411 |
| | <0.5 | 159 | 88 | 113 | 377 | 26 | 763 |
| | >0.7 | 88 | 13 | 112 | 111 | 13 | 337 |
| 22e gswp3-w5e5 | 0.5-0.7 | 113 | 25 | 81 | 191 | 7 | 417 |
| | <0.5 | 159 | 88 | 115 | 367 | 26 | 755 |
| | >0.7 | 51 | 3 | 47 | 146 | 6 | 253 |
| 22e 20crv3-era5 | 0.5-0.7 | 91 | 19 | 79 | 151 | 4 | 344 |
| | <0.5 | 206 | 95 | 195 | 398 | 18 | 912 |
| | >0.7 | 56 | 3 | 50 | 123 | 16 | 248 |
| 22e 20crv3-w5e5 | 0.5-0.7 | 92 | 18 | 66 | 159 | 3 | 338 |
| | <0.5 | 209 | 107 | 194 | 393 | 20 | 923 |
| | >0.7 | 77 | 6 | 103 | 127 | 7 | 320 |
| 22e gswp3-era5 | 0.5-0.7 | 113 | 21 | 99 | 176 | 6 | 415 |
| | <0.5 | 160 | 88 | 122 | 387 | 17 | 774 |

**Table D2.** Model performance and the $KGE_r$ efficiency indicator and number of basins per Köppen-Geiger region in the particular performance class for the different WaterGAP variants.

| Model variant | $KGE_r$ | A | B | C | D | E | sum |
|---|---|---|---|---|---|---|---|
| | >0.8 | 210 | 31 | 186 | 231 | 18 | 676 |
| 22d gswp3-w5e5 | 0.5-0.8 | 120 | 53 | 99 | 258 | 17 | 547 |
| | <0.5 | 30 | 42 | 23 | 180 | 11 | 286 |
| | >0.8 | 210 | 31 | 185 | 233 | 19 | 678 |
| 22e gswp3-w5e5 | 0.5-0.8 | 121 | 54 | 101 | 256 | 15 | 547 |
| | <0.5 | 29 | 41 | 22 | 180 | 12 | 284 |
| | >0.8 | 123 | 11 | 111 | 262 | 11 | 518 |
| 22e 20crv3-era5 | 0.5-0.8 | 182 | 57 | 156 | 246 | 13 | 654 |
| | <0.5 | 43 | 49 | 54 | 187 | 4 | 337 |
| | >0.8 | 141 | 12 | 116 | 228 | 20 | 517 |
| 22e 20crv3-w5e5 | 0.5-0.8 | 171 | 56 | 148 | 246 | 8 | 629 |
| | <0.5 | 45 | 60 | 46 | 201 | 11 | 363 |
| | >0.8 | 181 | 18 | 180 | 257 | 14 | 650 |
| 22e gswp3-era5 | 0.5-0.8 | 137 | 58 | 121 | 268 | 11 | 595 |
| | <0.5 | 32 | 39 | 23 | 165 | 5 | 264 |

**Table D3.** Model performance and the $KGE_b$ efficiency indicator and number of basins per Köppen-Geiger region in the particular performance class for the different WaterGAP variants.

| Model variant | $KGE_b$ | A | B | C | D | E | sum |
|---|---|---|---|---|---|---|---|
| | >1.5 | 0 | 4 | 0 | 1 | 0 | 5 |
| | 1.1-1.5 | 104 | 32 | 59 | 80 | 1 | 276 |
| 22d gswp3-w5e5 | 0.9-1.1 | 241 | 60 | 218 | 484 | 29 | 1032 |
| | 0.5-0.9 | 14 | 29 | 28 | 104 | 16 | 191 |
| | <0.5 | 1 | 1 | 3 | 0 | 0 | 5 |
| | >1.5 | 1 | 4 | 0 | 1 | 0 | 6 |
| | 1.1-1.5 | 96 | 33 | 56 | 89 | 2 | 276 |
| 22e gswp3-w5e5 | 0.9-1.1 | 249 | 58 | 222 | 484 | 28 | 1041 |
| | 0.5-0.9 | 13 | 30 | 27 | 95 | 16 | 181 |
| | <0.5 | 1 | 1 | 3 | 0 | 0 | 5 |
| | >1.5 | 0 | 4 | 4 | 5 | 0 | 13 |
| | 1.1-1.5 | 76 | 25 | 97 | 99 | 8 | 305 |
| 22e 20crv3-era5 | 0.9-1.1 | 246 | 53 | 190 | 540 | 20 | 1049 |
| | 0.5-0.9 | 26 | 30 | 28 | 50 | 0 | 134 |
| | <0.5 | 0 | 5 | 2 | 1 | 0 | 8 |
| | >1.5 | 0 | 4 | 5 | 4 | 0 | 13 |
| | 1.1-1.5 | 86 | 35 | 88 | 96 | 3 | 308 |
| 22e 20crv3-w5e5 | 0.9-1.1 | 251 | 63 | 184 | 481 | 24 | 1003 |
| | 0.5-0.9 | 20 | 25 | 30 | 94 | 12 | 181 |
| | <0.5 | 0 | 1 | 3 | 0 | 0 | 4 |
| | >1.5 | 0 | 4 | 0 | 0 | 0 | 4 |
| | 1.1-1.5 | 94 | 19 | 68 | 93 | 10 | 284 |
| 22e gswp3-era5 | 0.9-1.1 | 232 | 61 | 224 | 540 | 18 | 1075 |
| | 0.5-0.9 | 23 | 26 | 30 | 56 | 2 | 137 |
| | <0.5 | 1 | 5 | 2 | 1 | 0 | 9 |

**Table D4.** Model performance and the $KGE_g$ efficiency indicator and number of basins per Köppen-Geiger region in the particular performance class for the different WaterGAP variants.

| Model variant | $KGE_g$ | A | B | C | D | E | sum |
|---|---|---|---|---|---|---|---|
| | >1.5 | 56 | 19 | 32 | 30 | 3 | 140 |
| | 1.1-1.5 | 68 | 21 | 69 | 57 | 8 | 223 |
| 22d gswp3-w5e5 | 0.9-1.1 | 68 | 18 | 110 | 109 | 9 | 314 |
| | 0.5-0.9 | 150 | 54 | 89 | 317 | 14 | 624 |
| | <0.5 | 18 | 14 | 8 | 156 | 12 | 208 |
| | >1.5 | 54 | 19 | 32 | 30 | 3 | 138 |
| | 1.1-1.5 | 70 | 23 | 70 | 57 | 8 | 228 |
| 22e gswp3-w5e5 | 0.9-1.1 | 67 | 17 | 110 | 107 | 8 | 309 |
| | 0.5-0.9 | 152 | 52 | 87 | 316 | 15 | 622 |
| | <0.5 | 17 | 15 | 9 | 159 | 12 | 212 |
| | >1.5 | 63 | 23 | 22 | 29 | 3 | 141 |
| | 1.1-1.5 | 40 | 12 | 67 | 79 | 9 | 207 |
| 22e 20crv3-era5 | 0.9-1.1 | 61 | 15 | 91 | 111 | 11 | 289 |
| | 0.5-0.9 | 165 | 57 | 127 | 294 | 3 | 646 |
| | <0.5 | 19 | 10 | 13 | 182 | 2 | 226 |
| | >1.5 | 65 | 23 | 33 | 32 | 2 | 155 |
| | 1.1-1.5 | 70 | 23 | 75 | 54 | 8 | 230 |
| 22e 20crv3-w5e5 | 0.9-1.1 | 61 | 24 | 106 | 100 | 10 | 301 |
| | 0.5-0.9 | 147 | 47 | 88 | 328 | 8 | 618 |
| | <0.5 | 14 | 11 | 8 | 161 | 11 | 205 |
| | >1.5 | 50 | 18 | 28 | 26 | 3 | 125 |
| | 1.1-1.5 | 42 | 10 | 70 | 77 | 7 | 206 |
| 22e gswp3-era5 | 0.9-1.1 | 50 | 10 | 89 | 121 | 12 | 282 |
| | 0.5-0.9 | 182 | 61 | 123 | 288 | 5 | 659 |
| | <0.5 | 26 | 16 | 14 | 178 | 3 | 237 |

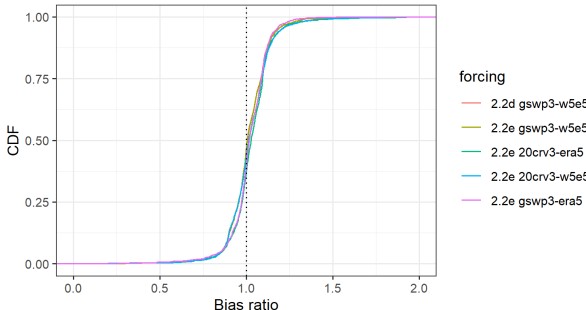

**Figure D3.** Cumulative distribution of the $KGE$ efficiency metric (bias parameter) for all streamflow values at the 1509 gauging stations for all model variants.

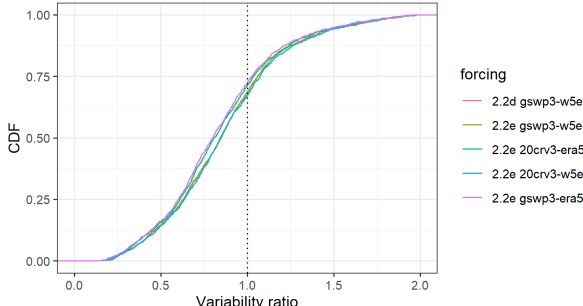

**Figure D4.** Cumulative distribution of the $KGE$ efficiency metric (variability parameter) for all streamflow values at the 1509 gauging stations for all model variants.

## Appendix E:  Standard model outputs

**Table E1.** Standard WaterGAP output variables: 1) Water storages. Units are $\mathrm{kg\,m^{-2}}$ (mm e.w.h.). All water storages except the reservoirstor are also available in a naturalized variant, indicated by the addend "nat" to the file. Temporal resolution is monthly except for two climate forcings that are additionally available in daily resolution.

| Storage type | GUDe variable file | Symbol in Müller Schmied et al. (2021) |
|---|---|---|
| Total water storage[1,2] | tws | $S_{tws}$ |
| Canopy water storage | canopystor | $S_c$ |
| Snow water storage | swe | $S_{sn}$ |
| Soil water storage | soilmoist | $S_s$ |
| Groundwater storage[2] | groundwstor | $S_g$ |
| Local lake storage[2] | loclakestor | $S_{ll}$ |
| Global lake storage[2] | glolakestor | $S_{lg}$ |
| Local wetland storage | locwetlandstor | $S_{wl}$ |
| Global wetland storage | glowetlandstor | $S_{wg}$ |
| Reservoir storage | reservoirstor | $S_{res}$ |
| River storage | riverstor | $S_r$ |

[1] Sum of all compartments below

[2] relative water storages, only anomalies with respect to a reference period can be evaluated

**Table E2.** Standard WaterGAP output variables: 2) Flows. Units are $\mathrm{kg\,m^{-2}\,s^{-1}}$ (mm e.w.h. $\mathrm{s^{-1}}$), except $\mathrm{m^3\,s^{-1}}$ for dis as well as K for triver. Temporal resolution is monthly.

| Flow type | GUDe variable file | Symbol in Müller Schmied et al. (2021) |
|---|---|---|
| Monthly precipitation | precmon | $P$ |
| Fast surface and fast subsurface runoff[1] | qs | $R_s$; $R_3$ in corrigendum |
| Diffuse groundwater recharge | qrdif | $R_g$ |
| Groundwater recharge from surface water bodies | qrswb | $R_{g_{l,res,w}}$ |
| Total groundwater recharge[2] | qr | $R_{g_{tot}}$ |
| Runoff from land[3] | ql | $R_l$ in corrigendum |
| Groundwater discharge[4] | qg | $Q_g$ |
| Total runoff from land[5] | qtot | sum of $Q_g$ and $R_s$ |
| Actual evapotranspiration [6] | evap | $E_a$ |
| Potential evapotranspiration | potevap | $E_p$ |
| Net cell runoff | ncrun | $R_{nc}$ |
| Streamflow[7] | dis | $Q_{r,out}$ |
| River water temperature | triver | n/a |

[1] fraction of total runoff from land that does not recharge the groundwater; [2] sum of qrdif and qrswb; [3] sum of qs and qrdif; [4] groundwater runoff; [4] sum of ql and qg; [6] sum of soil evapotranspiration, sublimation, evaporation from canopy, evaporation from water bodies and actual consumptive water use; [7] river discharge

**Table E3.** Standard WaterGAP output variables: 3) Water use. Units are $\mathrm{kg\,m^{-2}\,s^{-1}}$ (mm e.w.h. s$^{-1}$). Temporal resolution is monthly.

| Flow type | GUDe variable file | Symbol in Müller Schmied et al. (2021) |
|---|---|---|
| Potential consumptive water use for domestic sector | pdomuse | |
| Potential withdrawal water use for domestic sector | pdomww | |
| Potential consumptive water use for thermoelectric sector | pelecuse | |
| Potential withdrawal water use for thermoelectric sector | pelecww | |
| Potential consumptive water use for irrigation sector | pirruse | |
| Potential withdrawal water use for irrigation sector | pirrww | |
| Potential withdrawal water use for irrigation sector from groundwater resources | pirrwwgw | |
| Potential consumptive water use for livestock sector [1] | plivuse | |
| Potential consumptive water use for manufacturing sector | pmanuse | |
| Potential consumptive water use for manufacturing sector from groundwater resources | pmanusegw | |
| Potential withdrawal water use for manufacturing sector | pmanww | |
| Potential withdrawal water use for manufacturing sector from groundwater resources | pmanwwgw | |
| Potential net abstraction from surface water | pnas | |
| Potential net abstraction from groundwater | pnag | |
| Potential consumptive water use from groundwater | pgwuse | |
| Potential withdrawal water use from groundwater | pgwww | |
| Potential consumptive water use [2] | ptotuse | |
| Potential withdrawal water use [3] | ptotww | |
| Actual net abstraction from surface water | anas | $NA_s$ |
| Actual net abstraction from groundwater | anag | $NA_g$ |
| Actual consumptive water use [4] | atotuse | $WC_a$ |

[1] equals withdrawal water use; [2] sum of pnas and pnag; [3] sum of pdomww, pelecww, pirrww, plivuse, pmnww; [4] sum of anas and anag

**Table E4.** Standard WaterGAP output variables: 4) Additional files provided for a better understanding of the model outputs.

| Storage type | GUDe variable file | Symbol in Müller Schmied et al. (2021) |
|---|---|---|
| Calibration status of the basin | calstatus | $CS$ |
| Area correction factor from calibration | cfa | $CFA$ |
| Station correction factor from calibration | cfs | $CFS$ |
| Gamma factor from calibration | gamma | $\gamma$ |
| Continental area of the grid cell | continentalarea | |
| Flow direction in D8 schema | flowdirection | |
| Outflow cells to Oceans and inland sinks | outflowcells | |
| Rooting depth of the grid cell | rootdepth | |
| Maximum soil water capacity of the soil compartment | smax | |
| Commissioning year of the reservoirs | startyear | |

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
