# Peer review of "The global water resources and use model WaterGAP v2.2e: description and evaluation of modifications and new features"

_Geoscientific Model Development, 2023_

## Author Comment (AC1)

**Reply on RC1**

**R1**: This paper describes recent updates to the WaterGAP global water resources model (v2.2e) and provides benchmarks against observations.

As with previous versions, the model is impressive and the analysis is comprehensive. However, the improvements described in this paper, and their impacts on performance, appear fairly minor. There should be space in GMD for efforts that focus primarily on software and data updates rather than scientific ones, but revisions are needed.

**Answer**: Thank you for your time to review the manuscript and for providing your comments and suggestions. We will reply to the referee comment, indicated by **R1** by our answer, indicated by **Answer**, and corresponding actions, indicated by **Action** and textual changes in *italic font*:

**R1**: 1. The improvements include the treatment of reservoirs, updates to datasets for reservoirs, non-irrigation water use, and streamflow stations for calibration. Updates to model capabilities include PET representations, glaciers, and water temperature, though the latter has already been described in a previous publication. While all of these changes are justified, and no doubt an intensive effort, their impact on the modeled global water balance and distribution of NSE presented in the results does not seem to be a large change from previous versions of WaterGAP. Section 7.5.1 reports a nearly identical performance to the previous version. The scientific contribution of the new model capabilities should be more strongly justified.

**Answer**: Thank you for your suggestion. As mentioned by the referee in the statement of the beginning, GMD has space for such rather technical descriptions. In particular, the manuscript is submitted as a model description paper type which (among other purposes) "… should be detailed, complete, rigorous, and accessible to a wide community of geoscientists." (https://www.geoscientific-model-development.net/about/manuscript_types.html#item1). Our approach is to describe this model version, provide evaluation details and give examples of model output. Scientific contributions are not necessarily the focus of this manuscript type and not necessarily intended by this manuscript. It is rather a documentation for the model outputs e.g. used within the ISIMIP phase 3 and to bring together all the features of this model version in a description of the model version. So, we do not aim for more scientific justification as this is not the focus of this manuscript type.

**Action**: none

**R1**: 2. A main focus of the updates is the reservoir model. From the previous paper (2021), the release policy is assumed to follow Hanasaki (2006) and Döll (2009), which distinguishes between irrigation and non-irrigation reservoirs. This is a simplified rule that can be applied globally, but is often inaccurate at the level of individual reservoirs. The current paper does not investigate changes to this assumption, but removes a previous limitation about the maximum storage capacity for flood prevention. The accuracy of reservoir storage shown in Supplemental Figures S1, S2 from version 2.2c leaves much room for improvement, and it is not clear that removing the storage capacity threshold will fix this. The updated results after the change are not shown.

**Answer**: We have to apologize. The storage capacity threshold was already removed for the model version 2.2d (noted in Müller Schmied et al., 2021, Section A2.4 (last bullet point) and

the inclusion in this manuscript was a result of an internal communication problem. We agree that the generic reservoir algorithm has several limitations and efforts are ongoing to improve the representation of reservoirs in large-scale hydrological models (Dong et al, 2023, Otta et al, 2023, Shrestha et al, 2024, Steyard & Condon, 2024).**Action**: We have removed the update of the reservoir algorithm completely from this manuscript.

**R1**: 3. More information should be included about the potential scale mismatch between reservoir outflow (a point) mapped to a larger grid cell. The same goes for the stream gage data used for calibration (Step 3 in Section 2.5.2). It is possible this information is included in previous papers, but it would help to discuss here the potential impacts of this scale mismatch.

**Answer**: Indeed, there is a scale mismatch of a point information that is located somewhere in a grid cell and a drainage direction map at 0.5 x 0.5 degree spatial resolution. Basically, the model assumes that there is one river per grid cell, and generally, the whole grid cell(s) contribute to the runoff for the basin. With regards to both, the reservoir outflow location and the location of streamflow station for calibration, the given coordinate does not always fit to the hydrological situation and the (not always provided) upstream drainage area from the station/reservoir location and the drainage network. We have manually checked the location of the coordinate of the station/reservoir and its hydrological situation, esp. if the station or a reservoir outflow is located at a tributary or the main stream. Hence, we have co-registered stations to the best-located grid cell (in order of good match of observed and DDM upstream basin area) but decided not to use correction factors in case basin sizes differ (given all the other uncertainties and as we anyhow use a threshold of 9000 km2 ~ 4 grid cells as minimum for calibration). In the shapefiles of the calibration data (Müller Schmied & Schiebener 2022), the basin area from the data provider as well as from the drainage direction map are provided. We will not add this information to the manuscript. However, we have added some text to reflect this discussion.

**Action**: We have added the following text to Line 66 (after the citations Döll and Lehner, 2002 and Schewe and Müller Schmied, 2022): *The location of the new reservoirs was manually co-registered in the drainage network with the help of web-based map information in order to match the given hydrological situation, in particular if a reservoir is located on the mainstream or its tributary.*

We have added the following text to Line 143 (after "re-map the station to a grid cell that fits with the drainage network"): *Re-mapping of the position focused on accurately relating the station either to the mainstream of the river or the tributary. A correcting factor for mismatches of drainage areas between the values provided by the station data producers and those calculated from the drainage direction map was not implemented but both areas can be found in the shapefiles of Müller Schmied & Schiebener (2022).*

**R1**: 4. The calibration process attempts to find an optimal value of gamma, the runoff coefficient, to align the modeled mean annual streamflow within either 1% or 10% of the observed. Failing this, additional correction factors are applied to the runoff. While I can appreciate the difficulty of calibrating a global model, this calibration setup would have several problems for a basin-scale study. The gamma parameter can compensate for any mass balance error without a physical relationship to the runoff curve shown in Figure 3 of the 2021 paper. The additional correction factors only worsen this problem, and many regions of the model rely on these (Fig 4 of the current paper). By calibrating to mean annual data, monthly dynamics could be lost, though the efficiency metrics reported in Figure 7 seem to be doing well at many stations.

This calibration approach may be standard for global models. But at the basin scale, we could expect to see more diagnostics applied to investigate whether the results are physically based, or to analyze how much of the calibration uncertainty comes from each component of the mass balance. The calibration is more of a bias correction that is not able to distinguish between the many degrees of freedom in the model.

**Answer**: We fully agree that this bias adjustment it is a rather simple approach and it is good to add the term bias adjustment into the text for clarification. More extended calibration approaches are of course available and tested also with WaterGAP (Döll et al., 2024, Hasan et al., 2023), but not yet applicable on global scale, and also not included in a standard version of the model. Indeed, the original idea was to reduce biases by calibrating the HBV beta (our gamma) to match mean observed annual streamflow for water resources assessment and in many cases the uncertainties are large enough that gamma alone is not enough to achieve the aim. With regards to the last sentence of the first paragraph (the effect on monthly dynamics when only the mean value is calibrated), we here refer to the abstract of Hunger and Döll (2008) where, it was stated that "other flow characteristics like low flow, inter-annual variability and seasonality, the deviation between simulated and observed values also decreases significantly, which, however, is mainly due to the better representation fo average discharge but not of variability". This is also reflected in relatively weak performance of the KGE variability parameter.

We also agree that further diagnostics are needed to elaborate on reasons for different model performance but this is outside of focus of this manuscript.

**Action**: We introduce the term bias adjustment in the paper for clarity. In the beginning of Section 5.2 (Line 314) we add the sentence: *"The calibration as implemented in the standard version of WaterGAP focuses on adjusting biases in a rather simple method. More comprehensive approaches are currently in development (Döll et al., 2024, Hasan et al., 2023) and might be used in future model versions."*

**References**

Dong, N., Yang, M., Wei, J., Arnault, J., Laux, P., Xu, S., et al. (2023). Toward improved parameterizations of reservoir operation in ungauged basins: A synergistic framework coupling satellite remote sensing, hydrologic modeling, and conceptual operation schemes. Water Resources Research, 59, e2022WR033026. https://doi. org/10.1029/2022WR033026

Döll, P., Hasan, H. M. M., Schulze, K., Gerdener, H., Börger, L., Shadkam, S., Ackermann, S., Hosseini-Moghari, S.-M., Müller Schmied, H., Güntner, A., and Kusche, J.: Leveraging multi-variable observations to reduce and quantify the output uncertainty of a global hydrological model: evaluation of three ensemble-based approaches for the Mississippi River basin, Hydrol. Earth Syst. Sci., 28, 2259–2295, https://doi.org/10.5194/hess-28-2259-2024, 2024.

Hasan, H. M. M., Döll, P., Hosseini-Moghari, S.-M., Papa, F., and Güntner, A.: The benefits and trade-offs of multi-variable calibration of WGHM in the Ganges and Brahmaputra basins, EGUsphere [preprint], https://doi.org/10.5194/egusphere-2023-2324, 2023.

Müller Schmied, H., & Schiebener, L. (2022). The global water resources and use model WaterGAP v2.2e: streamflow calibration and evaluation data basis (1.1) [Data set]. Zenodo. https://doi.org/10.5281/zenodo.7255968

Otta, K., Müller Schmied, H., Gosling, S. N., and Hanasaki, N.: Use of satellite remote sensing to validate reservoir operations in global hydrological models: a case study from the CONUS, Hydrol. Earth Syst. Sci. Discuss. [preprint], https://doi.org/10.5194/hess-2023-215, in review, 2023.

Shrestha, P. K., Samaniego, L., Rakovec, O., Kumar, R., Mi, C., Rinke, K., & Thober, S. (2024). Toward improved simulations of disruptive reservoirs in global hydrological modeling. Water Resources Research, 60, e2023WR035433. https://doi.org/10.1029/2023WR035433

Steyaert, J. C. and Condon, L. E.: Synthesis of historical reservoir operations from 1980 to 2020 for the evaluation of reservoir representation in large-scale hydrologic models, Hydrol. Earth Syst. Sci., 28, 1071–1088, https://doi.org/10.5194/hess-28-1071-2024, 2024.

---

## Author Comment (AC2)

**Reply on RC2**

**R2:** The manuscript presents the most recent version of the WaterGAP model (v2.2e). Refinements, new algorithms, and updated data for calibration are presented. The manuscript is well-written, and the overall evaluation section is comprehensive and competently performed. My primary concern is the link between the extensive software modifications and data increment and the overarching scientific goal of assessing global and regional water resources. Even though the modifications, data additions, and model evaluation are comprehensively described, their scientific significance is not sufficiently justified in the context of water resources assessment. In this sense, I consider revisions (mostly related to rephrasing and further discussions) to be needed before publication. Please see below for my specific comments:

**Answer**: Thank you for your time to review the manuscript and for providing your comments and suggestions. We will reply to the referee comment, indicated by **R2** by our answer, indicated by **Answer**, and corresponding actions, indicated by **Action** and textual changes in *italic font*:

**R2: Comment 1:** The authors cite three papers to introduce the WaterGAP model. However, definitions necessary to understand the implemented modifications are missing. As a reader, I spent extra time reviewing the referenced papers to understand the main changes. Examples of these definitions include "naturalized mode" (L45), "standard runs" (L46-L47), and "local lakes algorithm" (L46), to name some just in section 2.1. Introducing the model's main features might be better so readers don't need to jump between papers (see related comment 2). As for GMD guidelines, for model description papers, it should be possible for independent scientists to build a model that, while not necessarily numerically identical, will produce equivalent results. In the current manuscript stage, the last is very hard to achieve.

**Answer**: Thank you. We agree that it is better readable when the introduction and the general concept is extended. So, we will extend the introduction while trying to find a good balance of providing the basis for understanding the model but avoiding repetition to the 2.2d description paper. The reader needs to read at least the comprehensive WaterGAP 2.2d description. GMD intends to publish further developments of models, so it can be expected that one paper does not describe the whole model, which is anyhow not possible in the case of complex models; and already in the 2.2d paper we had to limit the description to a certain extent. We believe that if we would repeat the lengthy description of the 2.2d paper we would not only run into duplication issues for many chapters but might receive referee comments that we should only highlight the differences of the specific model version. So, we intend to rely on the concept on describing mainly changes to the 2.2d paper.

With regard on your last point – we agree that the optimal goal of such model description papers should be to build a similar model based on these descriptions. Anyhow, this is hard to realize with such complex software that is in development since nearly three decades with it's thousands of lines of code. In a 3-year funded project, the model is currently being rewritten and re-implemented in Python (https://hydrologyfrankfurt.github.io/ReWaterGAP/) and this is already a tough task for us as WaterGAP developers due to the complexity; so it is probably unrealistic (and not practical) to aim for a reproduction of the software based on a description paper. Hence, the focus is more on describing the rationale and background of the model components.

**Action**: We further motivate the manuscript in the introduction and add further description; see also our reply to your Comment 2 below. To better allow the entry points for readers without pre-knowledge of the 2.2d paper, we have clarified the terms "naturalized" and "standard" by adding descriptions accordingly. More specific, we rename "standard" to "ant" (where appropriate) and "naturalized" to "nat" to be fully consistent to Müller Schmied et al, 2021. With regards to the local lakes algorithm as mentioned by the referee, we add the corresponding description section of the 2.2d paper to the end of Line 46.

In Line 18 we will add: "*Please note that this paper is not a thorough description of the whole WaterGAP model, it highlights only the modifications. For a comprehensive overview, the reader is referred to Müller Schmied et al. (2021).*

*WaterGAP was developed to quantify global-scale water resources as well as water stress with focus on direct human impacts in terms of human water use and artificial reservoirs. The model framework (Fig. 1) consists of sectoral water use models that are linked in a submodel (GSWSUSE) to calculate potential net water abstractions from surface water as well from groundwater. This acts as an input for the WaterGAP Global Hydrology Model that calculates the water storages and fluxes as well as routes the streamflow to the basin outlet (Fig. 1). WaterGAP as described here operates with a spatial resolution of 0.5° x 0.5° and at daily time steps. The model can be run in a standard mode ("ant", including direct human impacts) and a variety of other variants in terms of human water use and reservoirs, e.g., a simulation of naturalized water flows and storages that would occur if there where neither human water use nor global artificial reservoirs/regulated lakes ("nat").*".

[Figure]

*Fig. 1: Schematics of the WaterGAP framework and the WaterGAP Global Hydrology Model (both taken from Müller Schmied et al., 2021) and summary of data updates, process updates and new algorithms.*

**R2: Comment 2:** The model modifications are widely discussed in Section 2. However, as a reader, it is hard to visualize the model as a whole and identify the modified components. A scheme presenting the model's overall structure with the modified stages highlighted might be helpful.

**Answer**: Thank you for this suggestion. We agree that showing the schemes is a good idea. However, the general structure has not changed as compared to the 2.2d paper, so we re-use these two figures (side-by-side) and highlight the new features in model version 2.2e in a summarized form inside this figure. We believe this also helps the reader for introduction (see our reply to your Comment 1).

**Action**: We add a new Fig. 1 to the introduction and describe the general concept (see above).

**R2: Comment 3:** Following an analysis of the simulated monthly time series of reservoir water storage to observations for 16 reservoirs in the United States, the authors decided to drop off the 85% maximum storage capacity assumption implemented in version 2.2d (Müller Schmied et al. 2021) (due to model underestimation in 11 of the analyzed cases). This implies that the decision to remove the original assumption is based on a local-based analysis where ~30% of the model results did not show underestimation issues. Thus, I am failing to see the reason for extrapolating these local results to the global implementation of the model. Furthermore, Figures S1 and S2 might indicate more pressing issues related to the model's accuracy (e.g., seasonality) that might not be improved by simply removing the assumption.

**Answer**: We have to apologize. The storage capacity threshold was already removed for the model version 2.2d (noted in Müller Schmied et al., 2021, Section A2.4 (last bullet point) and the inclusion in this manuscript was a result of an internal communication problem. We agree that the generic reservoir algorithm has several limitations and efforts are ongoing to improve the representation of reservoirs in large-scale hydrological models (Dong et al, 2023, Otta et al, 2023, Shrestha et al, 2024, Steyard & Condon, 2024).

**Action**: We have removed the update of the reservoir algorithm completely from this manuscript.

**R2: Comment 4**: Section 5 presents the effects of the model modifications on multiple areas and the impact of differential forcing. Overall, the effect of the implemented changes seems to be minor. However, I found it challenging to visualize the minor differences because most of the baseline results (from v2.2d) are presented in Supplementary Material. I would recommend summarizing the results as differences of v2.2e from the original implementation rather than presenting each version independently. Furthermore, this issue connects to my general comment about the lack of scientific justification for the model modifications. If model parametrizations and data changes lead to almost negligible changes, the scientific rationale for implementing them should be discussed further.

**Answer**: With regards to the general comment (scientific justification), please see our response to your Comment 1. Some standard model version updates were driven by ISIMIP3 requirements (e.g. inclusion of GRanD update and simulation of water temperature) and other updates serve to keep the model up-to-date (e.g. the spatio-temporal update of the calibration data basis). Overall it was rather expected that changes in model performance might not appear large if globally averaged but may be large for individual cells and basins. In particular for the standard version, not too many changes are expected. The more „severe" modifications

are not in the standard version. But we fully agree, it is hard to see differences especially from the global basin maps, and a direct comparison might help here.

**Action:** We created spatial maps that show differences of model performance on a basin level, for 2.2e vs. 2.2d. The motivation is to see, where the difference to the optimal value (in terms of model performance indicator) is reduced or increased between the two versions that are driven by the same climate forcing and calibration data. For all indicators we calculate this difference as the ratio of the absolute deviation from the optimal indicator value (here everywhere 1.0) as:

$$PR\_[IND] = \frac{|1.0 - IND_{2.2e}|}{|1.0 - IND_{2.2d}|}$$

where:

PR_[IND]: Performance ratio of the given indicator IND [-]

IND: indicator value (KGE and its components for streamflow; Amplitude ratio for TWSA and the ratio of Model divided by GRACE for TWSA trend) for the particular model version

The smaller the resulting PR_[IND] is, the better 2.2e is compared to 2.2d. For PR_[IND] values < 1.0, 2.2e performs better than 2.2d and vice versa. The closer PR is to 0, the better performs 2.2e against 2.2d.

[Figure]

*Fig. 2: Resulting PR of streamflow for the model version 2.2d and 2.2e as driven by gswp3-w5e5 for overall KGE (a), KGE beta (b), KGE correlation r (c) and KGE variability gamma (d). Bluish colours indicate that 2.2e is closer to the optimal parameter indicator value than 2.2d. Note that the calibration procedure forces KGE beta values to be close to the optimum value, hence the drastic colours are a result of only small differences to the optimum value.*

[Figure]

*Fig. 3: Resulting PR of TWSA for the model version 2.2d and 2.2e as driven by gswp3-w5e5 for the amplitude ratio (a), correlation ratio (b) and trend ratio (c). Bluish colours indicate that 2.2e is closer to the optimal parameter indicator value than 2.2d.*

Figs 2 and 3 will be included in Section 7.5.1 (WaterGAP 2.2e vs. WaterGAP 2.2d) with a brief description, the indicator description PR_[IND] added as Appendix C

**References**

Dong, N., Yang, M., Wei, J., Arnault, J., Laux, P., Xu, S., et al. (2023). Toward improved parameterizations of reservoir operation in ungauged basins: A synergistic framework coupling satellite remote sensing, hydrologic modeling, and conceptual operation schemes. Water Resources Research, 59, e2022WR033026. https://doi. org/10.1029/2022WR033026

Otta, K., Müller Schmied, H., Gosling, S. N., and Hanasaki, N.: Use of satellite remote sensing to validate reservoir operations in global hydrological models: a case study from the CONUS, Hydrol. Earth Syst. Sci. Discuss. [preprint], https://doi.org/10.5194/hess-2023-215, in review, 2023.

Shrestha, P. K., Samaniego, L., Rakovec, O., Kumar, R., Mi, C., Rinke, K., & Thober, S. (2024). Toward improved simulations of disruptive reservoirs in global hydrological modeling. Water Resources Research, 60, e2023WR035433. https://doi.org/10.1029/2023WR035433

Steyaert, J. C. and Condon, L. E.: Synthesis of historical reservoir operations from 1980 to 2020 for the evaluation of reservoir representation in large-scale hydrologic models, Hydrol. Earth Syst. Sci., 28, 1071–1088, https://doi.org/10.5194/hess-28-1071-2024, 2024.

---

## Author Comment (AC3)

**General reply**

In addition to the replies to the two referees, we would like to use the occasion to update the manuscript in two directions. Please note that this does not imply modifications in the content but updates the output data description and enriches visualization of model output.

1. Updated repositories for model output

As the year 2023 is now completed, we have used the chance to update GSWP3-ERA5 model output to include the year 2023. Hence, the two repositories (with and without direct human impacts) have been updated (values before 2023 are not changed) (Müller Schmied et al, 2024c,d). Furthermore, we received several requests to provide daily model output of water storage compartments. Hence, we provide this for the two climate forcings gswp3-era5 and gswp3-w5e5 (with direct human impact only) (Müller Schmied et al., 2024a,b). These datasets will be added to the assets, and for the GSWP3-ERA5 updates, this replace the provided asset data in the original submission. Due to the update of GSWP3-ERA5 we have updated the global-scale water balance component tables (Tables S4, S8) in the Supplement.

2. Inclusion of a WaterGAP WebApp

After submission of this manuscript we have been in collaboration with a geodata company (www.ageoce.com) who used the provided model output to create a WaterGAP WebApp (https://www.ageoce.com/en/apps/watergap/). We add this Website as asset and refer to it in the end of Sect. 8 (L 586) as: *"A spatial view for a range of model output is available in a Web-App (Attard 2023)".*

**References**

Attard, G.: WaterGAP explorer: https://www.ageoce.com/en/apps/watergap/, last access: 1 June 2024.

Müller Schmied, H., Trautmann, T., Ackermann, S., Cáceres, D., Flörke, M., Gerdener, H., Kynast, E., Peiris, T. A., Schiebener, L., Schumacher, M., and Döll, P.: The global water resources and use model WaterGAP v2.2e - daily water storage model output driven by gswp3-w5e5 and historical setup of direct human impacts, https://doi.org/10.25716/GUDE.0K77-CXAC, 5 April 2024a.

Müller Schmied, H., Trautmann, T., Ackermann, S., Cáceres, D., Flörke, M., Gerdener, H., Kynast, E., Peiris, T. A., Schiebener, L., Schumacher, M., and Döll, P.: The global water resources and use model WaterGAP v2.2e - daily water storage model output driven by gswp3-era5 and historical setup of direct human impacts, https://doi.org/10.25716/GUDE.17VN-ZP9G, 5 April 2024b.

Müller Schmied, H., Trautmann, T., Ackermann, S., Cáceres, D., Flörke, M., Gerdener, H., Kynast, E., Peiris, T. A., Schiebener, L., Schumacher, M., and Döll, P.: The global water resources and use model WaterGAP v2.2e - model output driven by gswp3-era5 and historical setup of direct human impacts, https://doi.org/10.25716/GUDE.14B4-KQ6R, 4 April 2024c.

Müller Schmied, H., Trautmann, T., Ackermann, S., Cáceres, D., Flörke, M., Gerdener, H., Kynast, E., Peiris, T. A., Schiebener, L., Schumacher, M., and Döll, P.: The global water resources and use model WaterGAP v2.2e - model output driven by gswp3-era5 and

neglecting direct human impacts, https://doi.org/10.25716/GUDE.1TWP-GNQP, 4 April 2024d.

---

## Author Response (AR2)

**Reply to EC**

**EC:** The authors have partially addressed the concerns of both reviewers. While the scope of the paper that focuses primarily in describing the model updates and providing evaluation results is appropriate for GMD, as it currently stands, the paper could strongly benefit from a stronger introduction and a discussion section. I provide below some context for why these two additions would help the paper:

**Reply**: Thank you for your assessment and the chance to revise the manuscript. We reply to the editor comments **EC** by our **Reply.**

**EC:** 1. The current introduction is very sparse and doesn't really set the stage for the paper. This is an opportunity to explain why the changes that are made to the model are made; if this is done and the modeling results still don't improve modeling results, at least a rationale for why the changes are made is clear. If not, it gives the appearance that random additions were made without a concrete plan for addressing model improvement or process representation. As it currently reads, there are only a few sentences here and there that address this need. If this were expanded, this would greatly benefit providing context for all the work that was done. The introduction in the 2021 paper was much stronger and gives context for the work.

**Reply:** Thank you very much for this clarification. We fully agree that such a rationale would help to set the scope of the paper but also to avoid the impression that the modifications are somehow random. We have strongly extended the introduction (from 2 to 4 pages) to provide the scientific rationale of all the modifications and additions in the new model version, also including more references to the literature.

**EC:** 2. The paper could benefit from a discussion section. The authors recognize issues of the calibration (among other issues) but fail to address them directly in the paper. Reviewer #1 provided feedback but that feedback was mostly deflected. Even if the weaknesses of the calibration (among others) are not addressed directly in an update to the methodology and results; it should be discussed more explicitly. More broadly, adding a discussion section that also provides a perspective on future efforts and model development focus in the future would be a valuable addition to the paper and would provide the additional information that provides stronger context for the results and future directions.

**Reply:** Thank you for the good idea to provide a discussion and future outlook section. As model calibration is one key element of the model, we have enhanced the results section by an assessment of the added value of the newly added 119 gauging stations (New section 7.5, new Fig. 13 and Table 10). Furthermore, we have included this topic in a new (discussion) section 8 on benefits and limitations of the calibration approach. Furthermore, we have improved the Conclusions section by an outlook that covers three topics: 1. Modification of the reservoir algorithm, 2. Inclusion of an updated glacier model, 3. Converting WaterGAP into a community-based model.

**Reply to RC1**

**RC1:** The authors have clarified that the goal of the paper is to provide a complete description of the current version of the model rather than a scientific contribution based on the model outputs. It may be a goal of GMD to provide space for such contributions. The authors have mostly deflected, suggesting that the model updates do not necessarily lead to a quantifiable improvement in terms of accuracy or process representation.

On the other points, the updated reservoir algorithm has been removed from the manuscript. The bias adjustment has been clarified, though remains simple and difficult to justify.

**Reply**: Thank you for your continued efforts in reviewing and commenting on the manuscript, we appreciate your time and thoughts. We reply to the referee comments **RC1** by indicated by **Reply**, and corresponding actions, indicated by **Action.**

**RC1:** It is true that the paper describes significant effort on the model development, and modeling at this scale is no small task. But the scientific justification appears to rest on previous studies, and the reader must take at face value the authors' claims of an improved model without much notable difference in the output. This reflects a broader problem with global models that are challenged by overparameterization and scale mismatches when accounting for human impacts on the water cycle.

**Reply:** Thank you for pointing out that the scientific justification is lacking. We would like to reply to this comment in three ways. 1. In order to frame the scientific scope, we have revised the introduction to provide a scientific background of the modifications in the model version. This also leads to scientific objectives that are written at the end of the introduction. 2. We have included a discussion of the calibration, in particular the benefit of the newly introduced stations for calibration. 3. Within the conclusions, we have integrated paragraphs about future perspectives on model development. We trust these modifications lead to a better scientific justification but also touches general issues with global-scale modelling.

**Action:** 1. Within the introduction, we describe the scientific reasons for each of the modifications. 2. At the end of the introduction we formulate objectives. 3. We introduce a new section „Discussion of calibration" that focuses on the calibration topic and the added value of the 119 newly introduced gauging stations for calibration. Furthermore, we further discuss the limitations of the calibration procedure and consequences when modifying maximum soil moisture, but also elaborate on alternative (multivariate) calibration procedures. 4. We extend the section „Conclusion" with an outlook of future model developments and provide three examples to improve global scale hydrological modeling.

**RC1:** The updates to the paper itself are fairly minor, so my concerns from the previous round remain. It would be difficult for the paper to move forward as a standard scientific contribution. However, I will leave it to the editors whether the technical description of the model improvements is sufficient to move forward in GMD.

**Reply:** We believe this comment is rather dedicated to the Editor and cannot add to our reply to the original comment 1 of Referee #1 (that it is within the scope of GMD, in particular of this manuscript type, to provide technical descriptions).